# DuoGPT: Training-free Dual Sparsity through Activation-aware Pruning in LLMs

**Ruokai Yin, Yuhang Li, Donghyun Lee, Priyadarshini Panda**
Yale University
ruokai.yin@yale.edu

## Abstract

Large language models (LLMs) deliver strong performance but are difficult to deploy due to high memory and compute costs. While pruning reduces these demands, most methods ignore activation sparsity observed at runtime. We reinterpret activation sparsity as dynamic structured weight sparsity and propose **DuoGPT**, a unified framework that constructs dual-sparse (spMspV) workloads by combining unstructured weight pruning with activation sparsity. To preserve accuracy, we extend the Optimal Brain Compression (OBC) framework with activation-aware calibration and introduce output residuals from the dense model as correction terms. We further optimize the solution for efficient GPU execution, enabling scalability to billion-parameter LLMs. Evaluations on LLaMA-2 and LLaMA-3 show that **DuoGPT** outperforms state-of-the-art structured pruning methods by up to 9.17% accuracy at an iso-speedup of $1.39\times$ compared to the baseline dense model. Code is available at GitHub.

## 1 Introduction

Large language models (LLMs) have made significant advances in performance across a range of complex, real-world language tasks (Brown et al. 2020, Touvron et al. 2023). However, their massive parameter counts present practical challenges for deploying these large pre-trained models during inference. For example, deploying LLaMA-2-70B (Touvron et al. 2023), an open-source pre-trained LLM with 70 billion parameters, typically requires 150 GB of GPU and CPU RAM, along with 60 GFLOPs of computation to decode a single token. Consequently, there is widespread interest in reducing both the storage and computational requirements of LLMs through techniques collectively referred to as model compression. Existing model compression techniques for LLMs include quantization (Frantar et al. 2022, Ashkboos et al. 2024b), tensor decomposition (Wang et al. 2024, Lin et al. 2024), and pruning (Frantar & Alistarh 2023, Ashkboos et al. 2024a). This work focuses on pruning, specifically incorporating activation sparsity into the one-shot weight pruning framework.

One-shot pruning methods selectively zero out the elements in the weight matrices of an LLM during one forward pass, and optionally update the remaining non-zero weights to compensate for the pruning error. Depending on the freedom in the zero patterns, the pruning can be further categorized into *unstructured pruning* and *structured pruning*. In general, the following tradeoff between the two categories has long stood: more structured sparsity yields more speedup[1], while more unstructured sparsity is associated with better accuracy performance. In this work, we strike a balance between the two by incorporating activation sparsity. We observe that during the decoding stage of LLMs (i.e., General Matrix-Vector (GEMV) operations[2]), activation sparsity can be interpreted as structured weight sparsity, as only weight rows corresponding to non-zero activations are involved

---

[1]We focus on the speedup of running LLMs on general purpose architectures, e.g., GPUs.
[2]We primarily focus on single-batch decoding in this work.

39th Conference on Neural Information Processing Systems (NeurIPS 2025).

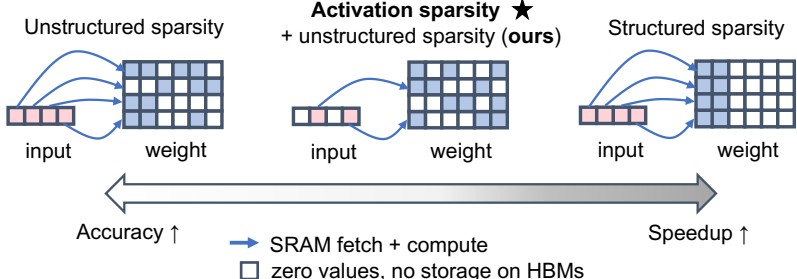

Figure 1: GEMV operation for the single-batch decoding stage under different types of sparsity.

in computation. Further, activation sparsity has been observed in most LLMs due to the non-linear activation functions (Liu et al. 2023, Li et al. 2022). Despite its acceleration potential, activation sparsity still requires storing the full dense model on the GPU, resulting in significant memory overhead. To reduce this burden, we propose leveraging activation sparsity on top of unstructured weight pruning. Our method thus produces dual-sparse LLM workloads that achieve a favorable balance between accuracy and efficiency, as illustrated in Figure 1.

Although combining dynamic activation sparsity with static weight sparsity is appealing, doing so naively can degrade performance. To address this, we extend the Optimal Brain Compression (OBC) framework (Frantar & Alistarh 2023) to compensate for runtime errors introduced by activation sparsity. Inspired by asymmetric calibration (Li et al. 2025), we reconstruct a layer-wise objective using the sparse activation from the pruned model and the corresponding dense activation from the original model. This enables weight updates during pruning calibration, mitigating accuracy degradation caused by sparse activations. Our contributions are as follows:

1. We reinterpret activation sparsity as a dynamic, structured form of weight sparsity. Combined with unstructured pruning, this perspective enables the construction of sparse matrix–sparse vector (spMspV) workloads for compressing and accelerating LLMs during decoding.

2. To compensate for the runtime errors introduced by dynamic activation sparsity, we propose **DuoGPT**, a layer-wise iterative pruning method that extends the OBC framework to activation sparsity-aware unstructured pruning in LLMs. With an efficient implementation, **DuoGPT** can calibrate a 70B-parameter LLaMA-3 model in under 130 minutes on a single A100 80GB GPU.

3. We conduct extensive experiments on LLaMA-2 and 3, demonstrating that **DuoGPT** significantly improves dual-sparse model performance across tasks. Under 50% dual weight-activation sparsity, **DuoGPT** outperforms state-of-the-art structured-pruning (e.g., Short-GPT (Men et al. 2024)) by 9.17% accuracy at an iso-speedup of roughly $1.4\times$ relative to the dense baseline. Compared to the sparse-activation (e.g., R-Sparse (Zhang et al. 2025)) methods, **DuoGPT** achieves up to a $1.97\times$ model size reduction at iso-accuracy.

## 2 Related Work

**Activation Sparsity.** Activation sparsity is a well-established property in ReLU-based LLMs (Li et al. 2022, Mirzadeh et al. 2023) and also emerges with newer activations like SwiGLU (Zhang et al. 2024a, Liu et al. 2023, Zhang et al. 2021, Haziza et al. 2025), naturally inducing sparsity in Multi-layer Perceptron (MLP) activations. This has motivated methods such as MoE-style MLP reconfiguration (Zhang et al. 2021, 2024b, Liu et al. 2023) and threshold-based activation pruning from calibration data (Lee et al. 2024a). Recent efforts have extended activation pruning to all transformer layers using thresholding (Liu et al. 2025) or hybrid SVD-based techniques (Zhang et al. 2025). Unlike these approaches—which retain dense weights—our work integrates activation sparsity with unstructured weight pruning, yielding a dual-sparse LLM workload. Notably, threshold-based techniques from prior work are complementary and can be incorporated into our framework. For reference, we also briefly compare with three recent activation sparsity works: TEAL (Liu et al. 2025), R-Sparse (Zhang et al. 2025), and CATS (Lee et al. 2024a), in Table 3b, Table 5, and Table 8.

**One-shot Weight Pruning for LLMs.** One-shot pruning, originally proposed by Han et al. (2015), removes weights based on magnitude but performs poorly at LLM scale (Frantar & Alistarh 2023).

SparseGPT (Frantar & Alistarh 2023) improves this with an approximate regression solver, enabling accurate pruning of 100B+ parameter models on a single GPU without fine-tuning. SlimGPT (Ling et al. 2024) extends this to structured pruning. Wanda (Sun et al. 2023) offers compensation-free unstructured pruning using the magnitudes of weights and activations. SliceGPT (Ashkboos et al. 2024a) prunes both weights and activations while preserving computational invariance, but suffers sharp accuracy drops beyond 30% sparsity. Other approaches prune at the block or submodule level using importance scores (Men et al. 2024, Zhong et al. 2024, Sandri et al. 2025). Different from recent work (Lee et al. 2024b) that combines the actual structured and unstructured pruning, our approach rather reinterprets activation sparsity as dynamic structured weight sparsity so that no actual structured pruning happens during calibration time.

**Optimal Brain Compression (OBC).** OBC builds on the Optimal Brain Surgeon (OBS) framework (LeCun et al. 1989, Hassibi & Stork 1992), which leverages second-order derivatives and the inverse Hessian to guide weight pruning. OBC (Frantar & Alistarh 2022) approximates this approach using an iterative solver, yielding closed-form solutions for optimal weight selection and updates. SparseGPT (Frantar & Alistarh 2023) was the first to scale this framework effectively to LLMs. Our work extends this line by incorporating activation sparsity into the OBC formulation and introducing a correction term derived from dense inputs. This design is inspired by the asymmetric calibration strategy recently proposed by Li et al. (2025) for OBC-based quantization.

## 3 Preliminaries

**Notations.** Throughout this paper, row vectors are denoted by bold lowercase letters and matrices by bold uppercase letters. For example, $\mathbf{w} \in \mathcal{R}^{1 \times k}$ represents a row vector of the weight. Consequently, the linear operation between the weight $\mathbf{W} \in \mathcal{R}^{n \times k}$ and the input calibration matrices $\mathbf{X} \in \mathcal{R}^{k \times m}$ is expressed as: $\mathbf{Y} = \mathbf{W}\mathbf{X}$[3]. Here, $k$ denotes the hidden dimension, $n$ is the number of output channels, and $m$ is the number of input tokens. Pruning is done through the element-wise multiplication, for example, $\hat{\mathbf{w}} = \mathbf{w} \odot \mathbf{m}^{\mathrm{w}}$ represents the pruning of one row of weight. Here, $\mathbf{m} \in \{0, 1\}$ is a bitmask. We use superscripts to specify different masks. Subscripts are used to specify the subsets of vectors and matrices. A negative subscript index on a matrix indicates the removal of the corresponding row and column. For example, $\mathbf{A}_{-j}$ is the matrix $\mathbf{A}$ with its $j^{\mathrm{th}}$ column and $j^{\mathrm{th}}$ row removed.

**Activation Sparsity.** We induce activation sparsity by element-wise multiplying the input activation $\mathbf{x}$ with a binary mask $\mathbf{m}^{\mathrm{x}}$, i.e., $\mathbf{x} \odot \mathbf{m}^{\mathrm{x}}$. The activation sparsity level $\mathrm{p}^{\mathrm{x}}$ is defined as the proportion of zeros in $\mathbf{m}^{\mathrm{x}}$. For a given linear layer $l$ during the decoding stage, the computation becomes: $\mathbf{y} = \mathbf{W}(\mathbf{x} \odot \mathbf{m}^{\mathrm{x}}) = \sum_{i \in \mathbf{m}_i^{\mathrm{x}}=1} \mathbf{W}_{:,i}\hat{\mathbf{x}}_i$. where $\hat{\mathbf{x}}_i$ denotes the sparse input vector. During calibration, when the input calibration data is in matrix form, we enforce that each column of the input has a sparsity level of $\mathrm{p}^{\mathrm{x}}$. Activations are pruned based on their absolute magnitude on the fly. In contrast to prior works (Liu et al. 2025, Zhang et al. 2025) that search for layer-specific optimal activation sparsity levels, we investigate a simpler setting in which uniform sparsity is applied across all transformer layers. At decoding time, our method is agnostic to how activation sparsity is induced (Details can be found in Section 5).

**OBC Framework for Pruning.** The OBC (Frantar & Alistarh 2022) pruning framework converts dense weight matrices $\mathbf{W}$ to a sparse one $\hat{\mathbf{W}}$ through a calibration process to preserve model performance. This calibration process is efficiently implemented by SparseGPT (Frantar & Alistarh 2023) as an iterative layer-wise pruning framework for LLMs. Formally, for each row of weight matrix, the calibration targets to minimize the differences between the original and pruned layer outputs: $\min_{\Delta \mathbf{w}} ||(\Delta \mathbf{w} + \mathbf{w})\mathbf{X} - \mathbf{w}\mathbf{X}||_F^2$, s.t. $\Delta \mathbf{w}\mathbf{e}_p^\top + \mathbf{w}_p = 0$, where $\mathbf{e}_p^\top$ is the unit vector that corresponds to $\mathbf{w}_p$, the weight that is removed. The loss for $\mathbf{w}_p$ and the updates to remaining weights are:

$$\mathcal{L}_p = \frac{\mathbf{w}_p^2}{\mathbf{H}_{pp}^{-1}}, \ \Delta \mathbf{w} = -\frac{\mathbf{w}_p}{\mathbf{H}_{pp}^{-1}} \cdot \mathbf{H}_{p,:}^{-1}. \tag{1}$$

Here $\mathbf{H}^{-1} = (\mathbf{X}\mathbf{X}^\top)^{-1}$ denotes the inverse Hessian matrix. The weights are sorted by their loss value and the weight with the lowest loss value is removed. Then, we compensate the remaining weights according to Equation 1. Once the $p^{\mathrm{th}}$ weight is removed, the inverse Hessian is updated

---

[3]We align our notation with prior work in our method description. In implementation or illustration, the weight matrices are stored in column-major format, as shown in Figure 1 and 2(a).

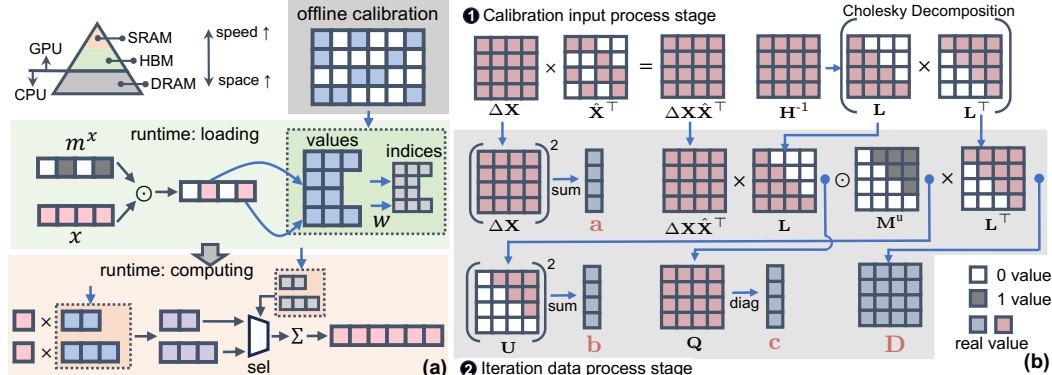

Figure 2: (a) Illustration of how dual-sparsity accelerates the decoding stage of LLMs by saving computation, memory loading, and storage. (b) Computing paradigm of the **DuoGPT**'s efficient GPU implementation. We neglect the element-wise division of $\text{diag}(\mathbf{L})$ for $\mathbf{c}$ in the figure.

through Gaussian elimination: $\mathbf{H}_{-p}^{-1} = (\mathbf{H}^{-1} - \frac{\mathbf{H}_{:,p}^{-1}\mathbf{H}_{p,:}^{-1}}{\mathbf{H}_{pp}^{-1}})$. This process iterates until the preset $\mathbf{p}^{\mathbf{w}}$ layer-wise weight sparsity is achieved. The resulting sparse model follows the closed-form solution.

# 4 DuoGPT

## 4.1 Activation Sparsity in Sparse LLMs

By leveraging activation sparsity, only certain rows of weights are fetched from High Bandwidth Memory (HBM) into Static RAM (SRAM) and further into the computation cores. However, since it is not known beforehand which specific rows will be required at runtime, memory storage cannot be reduced. As a result, the entire dense model must still be fully materialized in the GPU's HBM.

To address this limitation, we propose introducing sparsity into both activations and weights—a strategy referred to as ***dual-sparsity***. Specifically, we prune the dense model into a sparse version during an offline calibration process. During inference, only the compressed model parameters are loaded and transferred from CPU to GPU, significantly reducing both GPU HBM bandwidth and memory storage requirements. During linear layer computations, only the rows of weight matrices corresponding to non-zero activations are fetched from HBM into SRAM. Moreover, because each weight row is stored in compressed format, additional savings are achieved in SRAM loading, SRAM storage, and GEMM core computation. The overall runtime execution flow for this dual-sparse GEMV workload is illustrated in Figure 2(a).

A potential challenge with dual-sparsity lies in the overlap between activation and weight sparsity. Given activation and weight sparsity levels of $\mathbf{p}^{\mathbf{x}}$ and $\mathbf{p}^{\mathbf{w}}$, respectively, the fraction of weights loaded from HBM to SRAM could approach $(1-\mathbf{p}^{\mathbf{x}})$ in the worst case—particularly if the weight sparsity distribution is highly skewed. Empirically, however, we found that incorporating activation sparsity into the pruning calibration process helps distribute weight sparsity more uniformly across rows. For instance, for our calibrated LLaMA-2-7B model with $\mathbf{p}^{\mathbf{x}} = \mathbf{p}^{\mathbf{w}} = 0.5$, the maximum fraction of weights loaded from HBM to SRAM is approximately 0.25 (More details in Section 5 and Appendix). The remaining challenge is to determine ***how activation sparsity can be effectively integrated into the weight pruning calibration process***.

## 4.2 Activation Sparsity-Aware Pruning Calibration

A central challenge in leveraging activation sparsity for model pruning lies in incorporating this sparsity pattern directly into the calibration process. This section introduces a principled framework for solving this problem. Let $\mathbf{X}$ denote the output from the previously pruned layer during calibration. Applying magnitude-based pruning to $\mathbf{X}$ yields a sparse version of the calibration input, denoted as $\hat{\mathbf{X}}$, which maintains uniform sparsity $\mathbf{p}^{\mathbf{x}}$ across its columns. The calibration objective thus becomes minimizing the expression $||(\Delta\mathbf{w} + \mathbf{w})\hat{\mathbf{X}} - \mathbf{w}\hat{\mathbf{X}}||_F^2$. While utilizing sparse calibration data helps the model quickly adapt to activation sparsity during inference, it inevitably reduces the information

density available for accurately compensating the remaining unpruned weights. This fundamental trade-off necessitates a more sophisticated approach.

The solution lies in asymmetric calibration (Li et al. 2025), which incorporates both sparse and dense data to mitigate information loss. Given $\tilde{\mathbf{X}}$ as the dense output from the dense model, the target becomes $||(\Delta\mathbf{w} + \mathbf{w})\hat{\mathbf{X}} - \mathbf{w}\tilde{\mathbf{X}}||_F^2$. Setting $\mathbf{w}(\tilde{\mathbf{X}} - \hat{\mathbf{X}}) = \mathbf{r}$ as the output residual between dense and sparse data, the target simplifies to $||\Delta\mathbf{w}\hat{\mathbf{X}} - \mathbf{r}||_F^2$. With the constraint of $\Delta\mathbf{w}\mathbf{e}_p^\top + \mathbf{w}_p = 0$, the Lagrangian formulation becomes:

$$\min_{\lambda, \Delta\mathbf{w}} \mathcal{L}_p = ||\Delta\mathbf{w}\hat{\mathbf{X}} - \mathbf{r}||_F^2 + \lambda(\Delta\mathbf{w}\mathbf{e}_p^\top + \mathbf{w}_p). \tag{2}$$

Solving this Lagrangian with $\mathbf{H} = \hat{\mathbf{X}}\hat{\mathbf{X}}^\top$ yields the induced error and optimal weight updates for the $p$-th weight (if removed):

$$\mathcal{L}_p = \frac{\mathbf{w}_p^2}{\mathbf{H}_{pp}^{-1}} + \mathbf{r}\mathbf{r}^\top - \mathbf{r}\hat{\mathbf{X}}^\top\mathbf{H}_{-p}^{-1}\hat{\mathbf{X}}\mathbf{r}^\top + \frac{2\mathbf{w}_p}{\mathbf{H}_{pp}^{-1}}\mathbf{r}\hat{\mathbf{X}}^\top\mathbf{H}_{:,p}^{-1}, \ \ \Delta\mathbf{w} = -\frac{\mathbf{w}_p}{\mathbf{H}_{pp}^{-1}} \cdot \mathbf{H}_{p,:}^{-1} + \mathbf{r}\hat{\mathbf{X}}^\top\mathbf{H}_{-p}^{-1}. \tag{3}$$

Equation 3 defines the iterative framework, **DuoGPT**, which prunes and calibrates the model using a closed-form solution. For each row $\mathbf{w}$, the algorithm identifies the $p$-th weight for pruning by computing $p = \arg\min_p(\mathcal{L}_p)$, then calculates a compensation term $\Delta\mathbf{w}$ to update the remaining weights. This process repeats until the target pruning ratio $\mathtt{p}^{\mathtt{w}}$ is achieved.

Despite its theoretically sound approach to effectively incorporating activation sparsity ($\hat{\mathbf{X}}$) into the calibration process and compensating the associated information loss with dense model's information ($\tilde{\mathbf{X}}$), the direct implementation of this framework becomes computationally infeasible for large-scale LLMs with billions of parameters. Two critical bottlenecks emerge: (1) A separate Hessian inverse ($\mathbf{H}^{-1}$) must be maintained and updated for each weight row at every iteration (Frantar & Alistarh 2023). (2) Selecting which weight to prune requires computing the loss $\mathcal{L}$ in Equation 3 for every candidate weight, followed by recalculating the corresponding compensation term $\Delta\mathbf{w}$ after pruning.

The naive implementation results in a total computational complexity of $\mathcal{O}(nmk^2 + nk^3)$ for a single layer. In the context of LLM calibration, both $m$ and $k$ reach substantial values (e.g., $m = 128 \times 2048 = 262{,}144$ and $k = 4096$ for LLaMA-2-7B), far exceeding the practical limits of modern GPUs. An efficient implementation is therefore essential, as detailed in the following section.

### 4.3 Efficient DuoGPT Algorithm

To overcome these challenges, we first make a simplification: assume that all weights to be pruned have already been selected, i.e., a fixed pruning mask $\mathbf{M}^{\mathtt{w}}$ is provided. Under this assumption, the asynchronous Hessian issue can be addressed through Hessian synchronization and weight freezing, as introduced in SparseGPT (Frantar & Alistarh 2023). The key insight is to process all rows in parallel by iterating through columns in a fixed order, rather than strictly following the optimal pruning order. This synchronization enables all rows to share a common Hessian, requiring only a single copy of $\mathbf{H}^{-1}$ in GPU memory. Moreover, all Hessian updates can be precomputed in a single step using Cholesky decomposition, where $\mathbf{H}^{-1} = \mathbf{L}\mathbf{L}^\top$ and $\mathbf{L}$ is the lower-triangular Cholesky factor. This formulation resolves the asynchronous Hessian challenge with minimal computational overhead while enhancing numerical stability (Frantar & Alistarh 2023).

The critical remaining problem is: **how to determine the pruning mask $\mathbf{M}^{\mathtt{w}}$?** More precisely, a pruning score metric $\mathbf{S}$ is needed for all weights to derive the pruning mask, effectively decoupling mask selection from the pruning and calibration stages. This score must satisfy three essential criteria: (1) **Efficiency**: The score must be inexpensive to compute, expressible in vectorized form, and amenable to precomputation. (2) **Adaptivity**: It must incorporate update information across iterations to avoid the performance degradation associated with static pruning structures. (3) **Fidelity**: The score should closely approximate the original loss metric $\mathcal{L}$ to ensure that the final result remains a good approximation of the exact closed-form solution.

To begin, the pruning score $\mathbf{S}$ is constructed using the error $\mathcal{L}$ incurred when each weight is removed, as derived from Equation 3. For each weight column $p$, the pruning scores are defined as:

$$\mathbf{S}_{:,p} = \frac{\mathbf{W}_{:,p}^2}{\mathbf{H}_{pp}^{-1}} + \mathrm{diag}(\mathbf{R}\mathbf{R}^\top) - \mathrm{diag}(\mathbf{R}\hat{\mathbf{X}}^\top\mathbf{H}_{-p}^{-1}\hat{\mathbf{X}}\mathbf{R}^\top) + \frac{2\mathbf{W}_{:,p}}{\mathbf{H}_{pp}^{-1}}\mathbf{R}\hat{\mathbf{X}}^\top\mathbf{H}_{:,p}^{-1}, \tag{4}$$

where $\mathbf{R} = \mathbf{W}(\hat{\mathbf{X}} - \tilde{\mathbf{X}})$ denotes the output residual for the full weight matrix. An observation is that Equation 4 comprises orthogonal error terms capturing distinct aspects of the objective function's landscape. The first term, inherited from SparseGPT, quantifies the intrinsic information loss from removing each weight in isolation. The subsequent terms emerge from **DuoGPT** and operate in complementary subspaces. The second term captures the quadratic self-interaction of output residuals, representing errors from the sparse-dense activation discrepancy. The third term functions as a negative correction, quantifying how each weight's removal affects activation-sparsity-induced errors. The final term represents a cross-correlation between weight importance and activation-sparsity effects, capturing dependencies that enable adaptive compensation for dual-sparsity effects. Together, these terms form a holistic metric that precisely quantifies each weight's contribution to model performance in the dual-sparse regime.

An important observation emerges from Equation 4: although output residuals help compensate for information loss introduced by activation sparsity, they must be updated whenever weights are modified: $\mathbf{R} \leftarrow \mathbf{R} - \Delta\mathbf{W}(\tilde{\mathbf{X}} - \hat{\mathbf{X}})$. This requirement not only eliminates the possibility of precomputing pruning scores but introduces a computational cost of $\mathcal{O}(mnk)$ per column, rendering the naive approach prohibitively expensive. An opportunity for simplification comes from decomposing $\mathbf{R}$ into a sum of partial-sum matrices formed by the outer product between each weight column and the corresponding row of the input difference $\Delta\mathbf{X} = \tilde{\mathbf{X}} - \hat{\mathbf{X}}$. Specifically, the residual matrix can be approximated as $\mathbf{R} = \sum_{j=0}^{k} \mathbf{W}_{:,j}\Delta\mathbf{X}_{j,:}$. Since $\mathbf{R}$ quantifies the discrepancy between sparse and dense model outputs, it can be decomposed into contributions from individual weight-activation pairs. This decomposition transforms Equation 4 into:

$$\mathbf{S}_{:,p} = \frac{\mathbf{W}_{:,p}^2}{\mathbf{H}_{pp}^{-1}} + \mathbf{R}_p\mathbf{R}_p^\top - \mathbf{W}_{:,p}\Delta\mathbf{X}_{p,:}\hat{\mathbf{X}}^\top\mathbf{H}_{-p}^{-1}\hat{\mathbf{X}}\Delta\mathbf{X}_{p,:}^\top\mathbf{W}_{:,p}^\top + \frac{2\mathbf{W}_{:,p}}{\mathbf{H}_{pp}^{-1}}\mathbf{W}_{:,p}\Delta\mathbf{X}_{p,:}\hat{\mathbf{X}}^\top\mathbf{H}_{:,p}^{-1}, \quad (5)$$

where $\mathbf{R}_p = \mathbf{W}_{:,p}\Delta\mathbf{X}_{p,:}$. Furthermore, since the pruning process is independent across rows, the term $\mathbf{W}_{:,p}^2$ can be factored out from the last three terms in Equation 5. This results in the following expression for the pruning score:

$$\mathbf{S}_{:,p} = \mathbf{W}_{:,p}^2\Big(\frac{1}{\mathbf{H}_{pp}^{-1}} + \textcolor{red}{\Delta\mathbf{X}_{p,:}\Delta\mathbf{X}_{p,:}^\top} - \textcolor{red}{\Delta\mathbf{X}_{p,:}\hat{\mathbf{X}}^\top\mathbf{H}_{-p}^{-1}\hat{\mathbf{X}}\Delta\mathbf{X}_{p,:}^\top} + \frac{2}{\mathbf{H}_{pp}^{-1}}\textcolor{red}{\Delta\mathbf{X}_{p,:}\hat{\mathbf{X}}^\top\mathbf{H}_{:,p}^{-1}}\Big). \quad (6)$$

This reformulation reveals an important property: pruning scores now adaptively update through explicit contributions from previously processed weight columns. Simultaneously, output residual information is implicitly incorporated by fusing the $\mathbf{R}$ updates into Equation 6, allowing $\mathbf{R}$ to evolve across iterations without explicit recalculation.

The next challenge is to compute Equation 6 efficiently. Denoting the three new terms (annotated in red) as $\mathbf{a}_p = \Delta\mathbf{X}_{p,:}\Delta\mathbf{X}_{p,:}^\top$, $\mathbf{b}_p = \Delta\mathbf{X}_{p,:}\hat{\mathbf{X}}^\top\mathbf{H}_{-p}^{-1}\hat{\mathbf{X}}\Delta\mathbf{X}_{p,:}^\top$, and $\mathbf{c}_p = \Delta\mathbf{X}_{p,:}\hat{\mathbf{X}}^\top\mathbf{H}_{:,p}^{-1}/\mathbf{H}_{pp}^{-1}$, efficiently computing Equation 6 is now equivalent to efficiently computing $\mathbf{a}$, $\mathbf{b}$, and $\mathbf{c}$.

First, all elements of $\mathbf{a}$ can be precomputed as $\mathbf{a} = \text{diag}(\Delta\mathbf{X}\Delta\mathbf{X}^\top)$, leveraging the independence across rows of $\Delta\mathbf{X}$. Second, the term $\mathbf{b}_p$ can be reformulated by applying the Cholesky decomposition to $\mathbf{H}_{-p}^{-1}$, which yields $\mathbf{H}_{-p}^{-1} = \mathbf{L}_{p+1:,p+1:}\mathbf{L}_{p+1:,p+1:}^\top$. Consequently, $\mathbf{b}_p$ transforms into: $\Delta\mathbf{X}_{p,:}\hat{\mathbf{X}}^\top\mathbf{L}_{p+1:,p+1:}\mathbf{L}_{p+1:,p+1:}^\top\hat{\mathbf{X}}\Delta\mathbf{X}_{p,:}^\top$. Exploiting row independence again, the entire $\mathbf{b}$ can be precomputed as $\mathbf{b} = \text{diag}(\mathbf{U}\mathbf{U}^\top)$, where $\mathbf{U}_{p,:} := \Delta\mathbf{X}_{p,:}\hat{\mathbf{X}}^\top\mathbf{L}_{p+1:,p+1:}$.

Finally, by again leveraging the Cholesky decomposition, $\mathbf{H}_{:,p}^{-1}/\mathbf{H}_{pp}^{-1}$ can be easily represented by $\mathbf{L}_{:,p}/\mathbf{L}_{pp}$ (Frantar & Alistarh 2023, Frantar et al. 2022). Thus $\mathbf{c}$ can be precomputed as $\mathbf{c} = \text{diag}(\Delta\mathbf{X}\hat{\mathbf{X}}^\top\mathbf{L}) \oslash \text{diag}(\mathbf{L})$, where $\oslash$ denotes the element-wise division (Proofs in Appendix).

A further insight is that $\mathbf{U}$ contains components from both the current iteration (the $p$-th column) and the subsequent one (the $(p+1)$-th column). Therefore, $\mathbf{U}$ can be re-expressed as: $\mathbf{U} = \Big((\Delta\mathbf{X}\hat{\mathbf{X}}^\top\mathbf{L}) \odot \mathbf{M}^{\text{u}}\Big)$, where $\mathbf{M}^{\text{u}} \in \mathcal{R}^{k \times k}$ is a strictly upper-triangular masking matrix with ones above the diagonal.

This reformulation offers a significant computational advantage: the shared intermediate term $\mathbf{Q} = \Delta\mathbf{X}\hat{\mathbf{X}}^\top\mathbf{L}$ can be reused for computing both $\mathbf{b}$ and $\mathbf{c}$. Furthermore, $\mathbf{U}$ itself can be reused to precompute the new weight compensation terms in Equation 3, specifically the term $\mathbf{r}\hat{\mathbf{X}}^\top\mathbf{H}_{-p}^{-1}$. Denoting this quantity as $\mathbf{D}$, it can be precomputed as $\mathbf{D} = \mathbf{U}\mathbf{L}^\top$, as demonstrated in Li et al. (2025).

**Algorithm 1** The **DuoGPT** activation sparsity-aware pruning for one layer with target of $\mathbf{p}^{\mathtt{w}}$ unstructured sparsity. Given lazy batch and mask selection block size $B$, each consecutive $B$ columns will be $\mathbf{p}^{\mathtt{w}}$ sparse.

---

**Input:** Dense weight $\mathbf{W}$, sparse calibration input $\hat{\mathbf{X}}$ with $\mathbf{p}^{\mathtt{x}}$ sparsity, Dense input $\tilde{\mathbf{X}}$.
$\mathbf{H} \leftarrow \hat{\mathbf{X}}\hat{\mathbf{X}}^{\top}, \mathbf{L} = Inverse\_Cholesky(\mathbf{H}), \Delta\mathbf{X} \leftarrow \tilde{\mathbf{X}} - \hat{\mathbf{X}}$
$\mathbf{Q} \leftarrow \Delta\mathbf{X}\hat{\mathbf{X}}^{\top}\mathbf{L}, \mathbf{U} \leftarrow \mathbf{Q} \odot \mathbf{M_u}$
$\mathbf{a} \leftarrow \mathrm{diag}(\Delta\mathbf{X}\Delta\mathbf{X}^{\top}), \mathbf{b} \leftarrow \mathrm{diag}(\mathbf{U}\mathbf{U}^{\top}), \mathbf{c} \leftarrow \mathrm{diag}(\mathbf{Q}) \oslash \mathrm{diag}(\mathbf{L}), \mathbf{D} \leftarrow \mathbf{U}\mathbf{L}^{\top}$
$\mathbf{P} \leftarrow \mathbf{0}_{n\times k}, \mathbf{S} \leftarrow \mathbf{0}_{n\times B}, \mathbf{M}^{\mathtt{w}} \leftarrow \mathbf{1}_{n\times B}, \mathbf{E} \leftarrow \mathbf{0}_{n\times B}$
**for** $i = 0, B, 2B, \ldots$ **do**
  **for** $j = i, i+1, \ldots, i+B-1$ **do**
    **if** $j \bmod B = 0$ **then**
      $\mathbf{S}_{:,j:j+B} \leftarrow \mathbf{W}^2_{:,j:j+B} \odot \left(\mathbf{1}^{\top}(\frac{1}{\mathrm{diag}(\mathbf{L})^2_{j:j+B}} + \mathbf{a}_{j:j+B} - \mathbf{b}_{j:j+B} + 2\mathbf{c}_{j:j+B})\right)$
      $\mathbf{M}^{\mathtt{w}}_{:,j:j+B} \leftarrow$ 0-mask $\mathbf{p}^{\mathtt{w}}$ of weights $w_c \in \mathbf{W}_{:,j:j+B}$ with the lowest $\mathbf{S}_{:,j:j+B}$
    **end if**
    $\mathbf{P}_{:,j} \leftarrow \mathbf{W}_{:,j} \odot \mathbf{M}^{\mathtt{w}}_{:,j}$
    $\mathbf{E}_{:,j-i} \leftarrow \frac{(\mathbf{W}_{:,j} - \mathbf{P}_{:,j})}{\mathbf{L}_{jj}}$
    $\mathbf{W}_{:,j:(i+B)} \leftarrow \mathbf{W}_{:,j:(i+B)} - \mathbf{E}_{:,j-i}\mathbf{L}^{\top}_{j,j:(i+B)} + \mathbf{W}_{:,j}\mathbf{D}_{j,j:(i+B)}$
  **end for**
  $\mathbf{W}_{:,(i+B):} \leftarrow \mathbf{W}_{:,(i+B):} - \mathbf{E}\mathbf{L}^{\top}_{i:(i+B),(i+B):} + \mathbf{W}_{:,i:(i+B)}\mathbf{D}_{i:(i+B),(i+B):}$ // lazy-batch updates
**end for**
$\mathbf{W} \leftarrow \mathbf{W} \odot \mathbf{M}^{\mathtt{w}}$

---

The pruning score $\mathbf{S}$ for an arbitrary column $p$ is thus reformed into the following efficient form:

$$\mathbf{S}_{:,p} = \mathbf{W}^2_{:,p}\left(\frac{1}{\mathbf{H}^{-1}_{pp}} + \mathbf{a}_p - \mathbf{b}_p + 2\mathbf{c}_p\right). \tag{7}$$

All newly introduced terms are now computable via vectorized operations. Particularly, since both $\mathbf{a}$ and $\mathbf{b}$ are expressed as the diagonal of inner products between matrices, they can be computed using in-place elementwise square operations followed by column-wise summation. As a result, computing the pruning scores for an entire layer reduces to $\mathcal{O}(mk^2)$ complexity—reducing the runtime of the naive implementation by a factor of $\min(n, \frac{nk}{m})$.

Moreover, since this efficient reformulation of $\mathbf{S}$ derives directly from the original error metric $\mathcal{L}$—with the only approximation being the decomposition of $\mathbf{R}$—the pruning score remains a faithful surrogate to $\mathcal{L}$, ensuring a close approximation to the exact closed-form solution. The efficient implementation of **DuoGPT** is illustrated in Figure 2(b), with the complete algorithm presented in Algorithm 1. Full derivations are provided in the Appendix.

Compared to the naive implementation with complexity $\mathcal{O}(nmk^2 + nk^3)$, **DuoGPT**'s new complexity is sufficient to make the algorithm practical, even for extremely large models. Additionally, SparseGPT's iterative blocking and lazy-batch updates are adopted to minimize data movement across iterations, further accelerating runtime. In practice, **DuoGPT** calibrates a LLaMA-3-70B model on a single 80GB A100 GPU in under 140 minutes.

### 4.4 Theoretical Analysis

We provide a brief theoretical analysis of **DuoGPT**, aiming to derive a lower bound on its guaranteed loss improvement over SparseGPT under activation-sparsity–aware pruning calibration. The complete proof is given in the Appendix, and our formulation builds upon the numerical error analysis framework in (Wu et al. 2016).

**Theorem 1.** *Under the activation sparsity level $\mathbf{p}^{\mathtt{x}}$ and given $\mathcal{L}_{SparseGPT}$ and $\mathcal{L}_{DuoGPT}$ as the loss functions of two methods during calibration. DuoGPT achieves the following guaranteed loss improvement over SparseGPT:*

$$\Delta\mathcal{L} = \mathcal{L}_{SparseGPT} - \mathcal{L}_{DuoGPT} = \mathbf{r}\hat{\mathbf{X}}^{\top}\mathbf{H}^{-1}_{-p}\hat{\mathbf{X}}\mathbf{r}^{\top} \geq \frac{\alpha\mathbf{p}^{\mathtt{x}}\sigma^2_r C^2_{\mathbf{w}}m}{\lambda_{max}(\mathbf{H})}, \tag{8}$$

*where $\alpha > 0$ is the stability constant measuring spectral gap preservation, $\lambda_{max}$ is the maximum eigenvalue of the Hessian $\mathbf{H} = \hat{\mathbf{X}}\hat{\mathbf{X}}^{\top}$, $\sigma_r > 0$ is the activation residual magnitude constant, $C_{\mathbf{w}}$ is the weight norm bound satisfying $||\mathbf{w}||^2_F \geq C_{\mathbf{w}}$, and $m$ is the calibration batch size.*

Theorem 1 predicts a consistent and interpretable improvement of **DuoGPT** over SparseGPT that scales linearly with activation sparsity level $p^x$. Experimentally, as shown in Tables 1, 3a, and 7, we observe consistent PPL improvement of **DuoGPT** across all model sizes and sparsity settings, validating the bound's predictive trend. Moreover, the empirical results reveal a clear scaling behavior: higher dual-sparsity levels yield larger performance gains, in agreement with the theoretical analysis.

## 5 Experiments

**Experimental Setup.** We implement **DuoGPT** using PyTorch (Paszke 2019) and the HuggingFace Transformer library (Wolf et al. 2019) for efficient model and dataset management. All calibration procedures and experiments are performed on 80GB NVIDIA A100 GPUs with offloading (two GPUs are specifically employed for zero-shot evaluations of 70B models). Unless otherwise stated, the calibration dataset consists of 128 2048-token samples, randomly selected from the C4 training dataset (Raffel et al. 2020). To assess model performance, we evaluate the perplexity (PPL) of our **DuoGPT**-pruned models on the WikiText2 dataset (Merity et al. 2016). Furthermore, we complement our evaluation by conducting 0-shot task classifications using the LM Eval Harness (Gao et al. 2021) across widely recognized downstream benchmarks: PIQA (Bisk et al. 2020), HellaSwag (Zellers et al. 2019), WinoGrande (Sakaguchi et al. 2021), ARC-easy, ARC-challenge (Clark et al. 2018), OpenBookQA (OBQA) (Mihaylov et al. 2018), and BoolQ (Clark et al. 2019). More detailed setups can be found in the Appendix.

**Unstructured Pruning Baselines Comparison.** We begin by comparing **DuoGPT** with dual-sparse baselines constructed using two widely adopted one-shot unstructured pruning methods: SparseGPT (Frantar & Alistarh 2023) and Wanda (Sun et al. 2023), as shown in Table 1. To ensure a fair comparison, we enable 50% activation sparsity for both **DuoGPT** and SparseGPT during calibration, allowing each method to perform activation-aware compensation. For Wanda, which lacks weight-adjustment compensation, the calibration data remains in dense format. All methods are evaluated with models pruned to 50% weight sparsity and run with 50% activation sparsity during inference. Additionally, we compare to the semi-unstructured (2:4) variants of these two approaches, which are widely regarded as achieving a balance between speedup and accuracy (Mishra et al. 2021). In the 2:4 setting, activations remain dense during both calibration and evaluation.

Table 1: Comparison against other one-shot unstructured pruning methods in their dual-sparse variants ($\mathbf{W}_{50\%} + \mathbf{X}_{50\%}$) and 2:4 variants ($\mathbf{W}_{2:4} + \mathbf{X}_{\text{dense}}$). **DuoGPT** has 50% dual-sparsity. We also report GPU hours required for calibration. Bold and underlined values indicate the best and second-best results, respectively.

| Model | Method | GPU Hours | Wiki2(↓) | PIQA | HellaSwag | ARC-Easy | ARC-Challenge | WinoGrande | Avg(↑) |
|---|---|---|---|---|---|---|---|---|---|
| LLaMA-3-8B | Dense | - | 6.14 | 80.63 | 79.13 | 77.53 | 53.33 | 72.93 | 72.71 |
| | SparseGPT | 0.21 | 14.05 | 72.96 | 61.97 | 62.29 | 37.80 | 62.51 | 59.51 |
| | SparseGPT (2:4) | 0.18 | 19.81 | 70.84 | 53.39 | 54.84 | 32.34 | 61.56 | 54.59 |
| | Wanda | 0.03 | 15.98 | 71.38 | 57.02 | 59.39 | 35.84 | 61.64 | 57.05 |
| | Wanda (2:4) | 0.03 | 25.13 | 67.85 | 47.97 | 50.80 | 29.95 | 58.96 | 51.11 |
| | **DuoGPT** | 0.27 | **13.41** | **73.72** | 61.89 | **63.01** | 36.77 | **64.80** | **60.04** |
| LLaMA-3-70B | Dense | - | 2.86 | 84.60 | 84.96 | 86.07 | 64.25 | 80.58 | 80.09 |
| | SparseGPT | 1.62 | 7.54 | 80.03 | 74.87 | **78.49** | 50.85 | 74.03 | 71.65 |
| | SparseGPT (2:4) | 1.50 | 13.32 | 77.53 | 63.78 | 72.64 | 46.33 | 72.77 | 66.61 |
| | Wanda | 0.26 | 8.19 | 80.03 | **76.90** | 77.10 | 50.09 | 70.17 | 70.86 |
| | Wanda (2:4) | 0.28 | 9.32 | 80.20 | 73.06 | 75.42 | 49.49 | 71.67 | 69.97 |
| | **DuoGPT** | 2.28 | **7.38** | **80.52** | 75.94 | 77.82 | **52.82** | **75.69** | **72.56** |
| LLaMA-2-7B | Dense | - | 5.47 | 79.11 | 75.99 | 74.58 | 46.25 | 69.06 | 69.00 |
| | SparseGPT | 0.18 | 8.98 | 73.88 | 63.87 | 62.67 | 35.75 | 63.38 | 60.02 |
| | SparseGPT (2:4) | 0.20 | 12.1 | 71.65 | 57.41 | 59.18 | 32.17 | 64.80 | 57.04 |
| | Wanda | 0.02 | 9.13 | 72.91 | 62.33 | 61.66 | 35.32 | 62.59 | 58.96 |
| | Wanda (2:4) | 0.04 | 12.2 | 70.95 | 54.94 | 57.28 | 31.40 | 62.27 | 55.37 |
| | **DuoGPT** | 0.22 | **8.58** | **73.88** | 63.75 | **64.18** | 35.67 | **65.11** | **60.52** |
| LLaMA-2-13B | Dense | - | 4.88 | 80.52 | 79.37 | 77.53 | 49.06 | 72.22 | 71.74 |
| | SparseGPT | 0.32 | 7.39 | 76.39 | 69.21 | 68.22 | 41.04 | 67.64 | 64.50 |
| | SparseGPT (2:4) | 0.33 | 9.56 | 74.16 | 62.95 | 63.09 | 37.46 | 66.22 | 60.78 |
| | Wanda | 0.06 | 7.41 | **76.39** | 69.19 | 67.51 | 40.02 | 65.90 | 63.80 |
| | Wanda (2:4) | 0.07 | 9.05 | 75.41 | 62.69 | 64.73 | 37.12 | 67.09 | 61.41 |
| | **DuoGPT** | 0.40 | **7.17** | 75.63 | 69.56 | **69.95** | 41.13 | 68.11 | **64.88** |
| LLaMA-2-70B | Dense | - | 3.32 | 82.75 | 83.81 | 81.02 | 57.34 | 77.90 | 76.56 |
| | SparseGPT | 1.68 | 5.09 | 79.54 | 77.51 | 78.87 | 52.73 | 75.37 | 72.80 |
| | SparseGPT (2:4) | 1.50 | 5.94 | 78.84 | 74.00 | 75.34 | 48.29 | 75.53 | 70.40 |
| | Wanda | 0.27 | 5.03 | **80.52** | 78.39 | 77.15 | **53.16** | 75.53 | 72.95 |
| | Wanda (2:4) | 0.30 | 5.47 | 79.71 | 76.01 | 77.36 | 50.77 | 75.53 | 71.88 |
| | **DuoGPT** | 2.30 | **5.02** | 80.41 | 77.64 | **78.96** | 52.56 | **77.51** | **73.42** |

Across all LLaMA models, **DuoGPT** consistently achieves the lowest perplexity, validating its capability to enhance pruned model performance under sparse activations. For example, on LLaMA-3-70B, **DuoGPT** reduces perplexity from 7.54 (SparseGPT) to 7.38. In zero-shot evaluations, it surpasses the second-best baseline results on WinoGrande by up to 2.14% and 2.29% for LLaMA-2 and 3, respectively. Overall, **DuoGPT** achieves the highest average accuracy across all evaluated tasks and models. Furthermore, compared to 2:4 weight-only sparse models, **DuoGPT** consistently yields improved performance. Notably, on LLaMA-3-8B, **DuoGPT** enhances the average accuracy from 54.59% (best-performing 2:4 baseline) to 60.04%, marking an improvement of 5.45%. We additionally report GPU hours (single A100 GPU with offloading) required for calibration. Owing to our efficient implementation detailed in Section 4.3, calibration of 70B models can be completed in approximately 2.3 hours.

Table 2: Compare with other structured pruning methods on LLaMA-2-7B. To align the normalized GPU speedup relative to the dense model, we set SliceGPT with 30% and other three baselines with 31.25% weight-only sparsity. **DuoGPT** has 50% dual-sparsity.

| Method | Model Size | Speedup | Wiki2(↓) | PIQA | HellaSwag | ARC-Easy | ARC-Challenge | WinoGrande | Avg(↑) |
|---|---|---|---|---|---|---|---|---|---|
| ShortGPT | 4.72B/6.74B | **1.44×** | 65.6 | 63.55 | 50.92 | 45.29 | 33.79 | 63.22 | 51.35 |
| 2SSP | 4.72B/6.74B | 1.31× | 12.4 | 71.71 | 63.11 | 55.05 | 31.83 | 64.17 | 57.34 |
| BlockPruner | 4.72B/6.74B | 1.41× | 20.5 | 69.53 | 54.77 | 51.89 | 32.00 | 60.85 | 53.81 |
| SliceGPT | 5.29B/6.74B | 1.26× | 23.1 | 71.22 | 58.04 | 56.14 | 33.11 | 64.88 | 56.68 |
| **DuoGPT** | **3.50B**/6.74B | 1.39× | **8.58** | **73.88** | **63.75** | **64.18** | **35.67** | **65.11** | **60.52** |

**Structured Pruning Baselines Comparison.** We compare our dual-sparse **DuoGPT** framework with several popular structured pruning methods: ShortGPT (Men et al. 2024), 2SSP (Sandri et al. 2025), BlockPruner (Zhong et al. 2024), and SliceGPT (Ashkboos et al. 2024a). All methods are evaluated using 256 samples of calibration data (32 for 2SSP, following its original setup). Each model is pruned to approximately 30% structured sparsity, resulting in up to 1.44× end-to-end speedup over the dense baseline. As shown in Table 2, by leveraging activation sparsity, **DuoGPT** achieves a competitive end-to-end speedup (1.39×) while maintaining significantly better accuracy compared to the structured pruning baselines. Additionally, **DuoGPT** yields the smallest model size post-compression, due to its higher 50% unstructured weight sparsity. If structured pruning methods were pushed to a comparable compression ratio of 50%, their performance would degrade sharply.

Table 3: Ablation studies of (a) different dual-sparsity levels and (b) other activation sparsity methods.

| Method | 30% | 40% | 55% | 60% | 65% |
|---|---|---|---|---|---|
| SparseGPT | 5.87 | 6.52 | 13.2 | 29.2 | 88.2 |
| Wanda | 5.85 | 6.49 | 16.0 | 67.1 | 248 |
| **DuoGPT** | **5.84** | **6.48** | **12.3** | **26.3** | **77.3** |

| Method | Model Size | OBQA | Arc-C | BoolQ | Avg |
|---|---|---|---|---|---|
| R-Sparse 50% | 6.90B/6.74B | 30.40 | 35.49 | **72.54** | 46.14 |
| **DuoGPT** + TEAL 50% | **3.50B**/6.74B | 38.40 | **36.77** | 71.77 | 48.98 |
| **DuoGPT** 50% | **3.50B**/6.74B | **39.00** | 35.67 | 72.48 | **49.05** |

(a) WikiText PPL of pruned LLaMA-2-7B models with varying dual-sparsity.

(b) Comparison and study of compatibility to other threshold-based activation sparsity work on LLaMA-2-7B models.

**Ablation Studies.** In Table 3a, we evaluate the performance of our method compared to other dual-sparse baselines across varying levels of dual-sparsity. As shown, **DuoGPT** consistently improves the performance of dual-sparse models at all sparsity levels. Notably, the performance gains become more pronounced as the dual-sparsity increases.

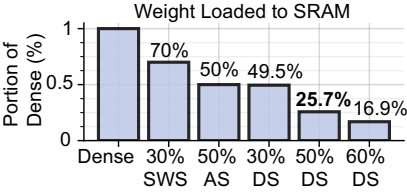

Figure 3: Number of weights of LLaMA-2-7B loaded to SRAM. SWS=structured weight sparsity, AS=activation sparsity, and DS=dual-sparsity.

We further investigate the compatibility of our method with threshold-based activation pruning during evaluation. As shown in Table 3b, applying the activation thresholding technique from TEAL (Liu et al. 2025) to a 50% dual-sparse **DuoGPT** model yields comparable performance to on-the-fly magnitude-based pruning. We also compare against R-Sparse (Zhang et al. 2025), a prior approach that combines activation sparsity with SVD-based low-rank weight compression. As Table 3b shows, **DuoGPT** achieves higher average accuracy than R-Sparse under 50% sparsity. Moreover, while our method reduces the model size to 50% of the dense baseline, R-Sparse incurs a 1% increase in memory usage due to the additional SVD branch.

Next, we analyze the number of weights loaded into SRAM across different sparsity paradigms, as shown in Figure 3. Consistent with our discussion in Section 4.1, we observe that the dual-sparse model loads approximately $(1-p^w) \times (1-p^x)$ fraction of the weights into SRAM. This result shows potential for further improving of the efficiency of dual-sparse LLMs with custom GPU spMspV kernels or ASIC dual-sparse accelerators.

We also demonstrate **DuoGPT**'s compatibility with weight-only quantization, a promising compression approach for accelerating memory-bounded LLM inference. We consider both 8-bit and 4-bit uniform integer quantization with channel-wise RTN (group size 128), which can be seamlessly integrated into our pruning framework (Frantar et al. 2022, Frantar & Alistarh 2022), and report the perplexity of both LLaMA-2-7B and LLaMA-2-13B on WikiText2 in Table 4.

Finally, we evaluate the performance of **DuoGPT** on a reasoning benchmark. Table 5 compares the accuracy of 50% activation-sparse TEAL, 30% dual-sparse SparseGPT, and **DuoGPT** on the 5-shot GSM8K (Cobbe et al. 2021) using LLaMA-3-8B, under the same number of SRAM weight fetches. **DuoGPT** achieves the highest accuracy and the smallest model size under iso-SRAM weight load.

Table 4: WikiText2 PPL with 50% dual-sparsity and varying weight precision.

| Model | Method | FP16 | INT8 | INT4 |
|---|---|---|---|---|
| LLaMA-2-7B | SparseGPT | 8.98 | 9.02 | 14.2 |
| | **DuoGPT** | **8.58** | **8.58** | **13.8** |
| LLaMA-2-13B | SparseGPT | 7.39 | 7.41 | 10.0 |
| | **DuoGPT** | **7.17** | **7.18** | **9.39** |

Table 5: Comparison with other baselines on 5-shot GSM8K with LLaMA-3-8B.

| Method $(X, W)_{sparsity}$ | Model Size | SRAM Loads | GSM8K |
|---|---|---|---|
| TEAL (50%, dense) | 8.03B/8.03B | 50.0% | 36.47 |
| SparseGPT (30%, 30%) | 5.94B/8.03B | 49.7% | 36.39 |
| **DuoGPT** (30%, 30%) | **5.94B**/8.03B | **49.6%** | **36.77** |

**Limitations and Future Work.** This work does not yet include a full GPU kernel implementation for accelerating the spMspV operation. Our current kernel implementation mainly exploits activation sparsity (spVM), which enabled the throughput measurements reported in our main paper. This demonstrates a lower bound on the real-world efficiency gains from our dual-sparsity approach. Unlike spMspM, spMspV avoids costly index matching, making it a strong candidate for efficient GPU acceleration—an avenue we leave for future exploration. While **DuoGPT** offers a promising approach to calibrate dual-sparse LLMs, this remains an open problem with substantial potential for further research.

Furthermore, our work focuses on inducing dual-sparsity only for linear layers. Other efforts exist to induce single-sided sparsity in attention layers. Integrating dual-sparsity with these recent works on both unstructured (Joo et al. 2025) and structured (Xu et al. 2024) KV cache pruning remains a relevant and promising direction.

# 6 Conclusion

In this paper, we present **DuoGPT**, an efficient one-shot dual-sparse pruning framework that unifies unstructured weight pruning with runtime activation sparsity. By extending the OBC framework with activation-aware calibration and residual correction, **DuoGPT** achieves state-of-the-art accuracy while significantly improving both memory and computational efficiency. Our method scales to large LLMs and complements existing activation pruning techniques, offering a practical path toward efficient and scalable LLM deployment.

# 7 Acknowledgement

This work was supported in part by CoCoSys, a JUMP2.0 center sponsored by DARPA and SRC, the National Science Foundation (CAREER Award, Grant #2312366, Grant #2318152), the DARPA Young Faculty Award and the DoE MMICC center SEA-CROGS (Award #DE-SC0023198).

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

# A Theoretical Derivation

## A.1 Proof of Equation 3 (Optimal Framework for Activation Sparsity-Aware Pruning)

We provide the detailed derivation of **DuoGPT**'s optimal calibration framework (Equation 3) proposed in Section 4.2.

*Proof.* The Lagrangian in Equation 2 is first expanded:

$$\mathcal{L} = \text{Tr}\left((\Delta \mathbf{w} \hat{\mathbf{X}} - \mathbf{r})^\top (\Delta \mathbf{w} \hat{\mathbf{X}} - \mathbf{r})\right) + \lambda \Delta \mathbf{w} \mathbf{e}_p^\top + \lambda \mathbf{w}_p$$
$$= \text{Tr}(\hat{\mathbf{X}}^\top \Delta \mathbf{w}^\top \Delta \mathbf{w} \hat{\mathbf{X}} - \hat{\mathbf{X}}^\top \Delta \mathbf{w}^\top \mathbf{r} - \mathbf{r}^\top \Delta \mathbf{w} \hat{\mathbf{X}} + \mathbf{r}^\top \mathbf{r}) + \lambda \Delta \mathbf{w} \mathbf{e}_p^\top + \lambda \mathbf{w}_p \tag{9}$$

To find the local minima of the Lagrangian in Equation 2, we set the partial derivatives with respect to $\Delta \mathbf{w}$ and $\lambda$ to zero. We first compute the partial derivative with respect to $\Delta \mathbf{w}$:

$$\frac{\partial \mathcal{L}}{\partial \Delta \mathbf{w}} = 2\Delta \mathbf{w} \hat{\mathbf{X}} \hat{\mathbf{X}}^\top - 2\mathbf{r} \hat{\mathbf{X}}^\top + \lambda \mathbf{e}_p$$
$$= 2\Delta \mathbf{w} \mathbf{H} - 2\mathbf{r} \hat{\mathbf{X}}^\top + \lambda \mathbf{e}_p, \tag{10}$$

and the partial derivative with respect to $\lambda$:

$$\frac{\partial \mathcal{L}}{\partial \lambda} = \Delta \mathbf{w} \mathbf{e}_p^\top + \mathbf{w}_p. \tag{11}$$

Setting the first equation to zero yields:

$$\Delta \mathbf{w} = \mathbf{r} \hat{\mathbf{X}}^\top \mathbf{H}^{-1} - \frac{\lambda}{2} \mathbf{e}_p \mathbf{H}^{-1}$$
$$= \mathbf{r} \hat{\mathbf{X}}^\top \mathbf{H}^{-1} - \frac{\lambda}{2} \mathbf{H}_{p,:}^{-1}. \tag{12}$$

Setting Equation 11 to zero and substituting $\Delta \mathbf{w}$ from Equation 12, we obtain:

$$\mathbf{r} \hat{\mathbf{X}}^\top \mathbf{H}^{-1} \mathbf{e}_p^\top - \frac{\lambda}{2} \mathbf{H}_{p,:}^{-1} \mathbf{e}_p^\top + \mathbf{w}_p = 0. \tag{13}$$

By simplifying with $\mathbf{H}_{p,:}^{-1} \mathbf{e}_p^\top = \mathbf{H}_{pp}^{-1}$, we solve for $\lambda$ as:

$$\lambda = \frac{2}{\mathbf{H}_{pp}^{-1}} (\mathbf{r} \hat{\mathbf{X}}^\top \mathbf{H}^{-1} \mathbf{e}_p^\top + \mathbf{w}_p)$$
$$= \frac{2}{\mathbf{H}_{pp}^{-1}} (\mathbf{r} \hat{\mathbf{X}}^\top \mathbf{H}_{:,p}^{-1} + \mathbf{w}_p) \tag{14}$$

Substituting Equation 14 back into Equation 12, we solve for $\Delta \mathbf{w}$ as:

$$\Delta \mathbf{w} = \mathbf{r} \hat{\mathbf{X}}^\top \mathbf{H}^{-1} - \frac{1}{\mathbf{H}_{pp}^{-1}} (\mathbf{r} \hat{\mathbf{X}}^\top \mathbf{H}_{:,p}^{-1} \mathbf{H}_{p,:}^{-1} + \mathbf{w}_p \mathbf{H}_{p,:}^{-1})$$
$$= -\frac{\mathbf{w}_p}{\mathbf{H}_{pp}^{-1}} \cdot \mathbf{H}_{p,:}^{-1} + \mathbf{r} \hat{\mathbf{X}}^\top (\mathbf{H}^{-1} - \frac{\mathbf{H}_{:,p}^{-1} \mathbf{H}_{p,:}^{-1}}{\mathbf{H}_{pp}^{-1}}). \tag{15}$$

By recognizing that the expression enclosed in parentheses in the second term corresponds to the Gaussian elimination operation introduced in Section 3, we obtain the final expression for $\Delta \mathbf{w}$:

$$\Delta \mathbf{w} = -\frac{\mathbf{w}_p}{\mathbf{H}_{pp}^{-1}} \cdot \mathbf{H}_{p,:}^{-1} + \mathbf{r} \hat{\mathbf{X}}^\top \mathbf{H}_{-p}^{-1}, \tag{16}$$

which is identical to the weight update term in Equation 3.

The next step is to derive the final expression for $\mathcal{L}$. Substituting Equation 16 into the loss function $\mathcal{L} = ||\Delta \mathbf{w} \hat{\mathbf{X}} - \mathbf{r}||_F^2$, the loss becomes:

$$\mathcal{L} = \left\| -\frac{\mathbf{w}_p}{\mathbf{H}_{pp}^{-1}} \mathbf{H}_{p,:}^{-1} \hat{\mathbf{X}} + \mathbf{r} \hat{\mathbf{X}}^\top \mathbf{H}_{-p}^{-1} \hat{\mathbf{X}} - \mathbf{r} \right\|_F^2. \tag{17}$$

This expression can be expanded as:

$$\mathcal{L} = \frac{\mathbf{w}_p^2}{(\mathbf{H}_{pp}^{-1})^2}\mathbf{H}_{p,:}^{-1}\mathbf{H}\mathbf{H}_{:,p}^{-1} - 2(\mathbf{r}\hat{\mathbf{X}}^\top\mathbf{H}_{-p}^{-1}\hat{\mathbf{X}} - \mathbf{r})(\frac{\mathbf{w}_p}{\mathbf{H}_{pp}^{-1}}\hat{\mathbf{X}}^\top\mathbf{H}_{:,p}^{-1}) + \mathbf{r}\hat{\mathbf{X}}^\top\mathbf{H}_{-p}^{-1}\mathbf{H}\mathbf{H}_{-p}^{-1}\hat{\mathbf{X}}\mathbf{r}^\top$$

$$- 2\mathbf{r}\hat{\mathbf{X}}^\top\mathbf{H}_{-p}^{-1}\hat{\mathbf{X}}\mathbf{r}^\top + \mathbf{r}\mathbf{r}^\top \quad (18)$$

$$= \frac{\mathbf{w}_p^2}{(\mathbf{H}_{pp}^{-1})^2}\mathbf{H}_{p,:}^{-1}\mathbf{H}\mathbf{H}_{:,p}^{-1} - 2\frac{\mathbf{w}_p}{\mathbf{H}_{pp}^{-1}}\mathbf{r}\hat{\mathbf{X}}^\top\mathbf{H}_{-p}^{-1}\mathbf{H}\mathbf{H}_{:,p}^{-1} + 2\frac{\mathbf{w}_p}{\mathbf{H}_{pp}^{-1}}\mathbf{r}\hat{\mathbf{X}}^\top\mathbf{H}_{:,p}^{-1}$$

$$+ \mathbf{r}\hat{\mathbf{X}}^\top\mathbf{H}_{-p}^{-1}\mathbf{H}\mathbf{H}_{-p}^{-1}\hat{\mathbf{X}}\mathbf{r}^\top - 2\mathbf{r}\hat{\mathbf{X}}^\top\mathbf{H}_{-p}^{-1}\hat{\mathbf{X}}\mathbf{r}^\top + \mathbf{r}\mathbf{r}^\top.$$

To simplify Equation 18, we observe that the second and third terms are nearly identical, differing only by the presence of the term $\mathbf{H}_{-p}^{-1}\mathbf{H}$. This same pattern appears between the fourth and fifth terms. We begin by simplifying $\mathbf{H}_{-p}^{-1}\mathbf{H}$:

$$\mathbf{H}_{-p}^{-1}\mathbf{H} = (\mathbf{H}^{-1} - \frac{\mathbf{H}_{:,p}^{-1}\mathbf{H}_{p,:}^{-1}}{\mathbf{H}_{pp}^{-1}})\mathbf{H}$$

$$= \mathbf{I} - \frac{\mathbf{H}^{-1}\mathbf{e}_p^\top\mathbf{e}_p\mathbf{H}^{-1}}{\mathbf{H}_{pp}^{-1}}\mathbf{H} \quad (19)$$

$$= \mathbf{I} - \frac{1}{\mathbf{H}_{pp}^{-1}}\mathbf{H}^{-1}\mathbf{e}_p^\top\mathbf{e}_p.$$

With the help of the identity matrix, we can more clearly see how the aforementioned nearly identical terms simplify. Substituting Equation 19 back into Equation 18:

$$\mathcal{L} = \frac{\mathbf{w}_p^2}{(\mathbf{H}_{pp}^{-1})^2}\mathbf{H}_{p,:}^{-1}\mathbf{H}\mathbf{H}_{:,p}^{-1} + 2\frac{\mathbf{w}_p}{(\mathbf{H}_{pp}^{-1})^2}\mathbf{r}\hat{\mathbf{X}}^\top\mathbf{H}^{-1}\mathbf{e}_p^\top\mathbf{e}_p\mathbf{H}_{:,p}^{-1}$$

$$- \frac{1}{\mathbf{H}_{pp}^{-1}}\mathbf{r}\hat{\mathbf{X}}^\top\mathbf{H}^{-1}\mathbf{e}_p^\top\mathbf{e}_p\mathbf{H}_{-p}^{-1}\hat{\mathbf{X}}\mathbf{r}^\top - \mathbf{r}\hat{\mathbf{X}}^\top\mathbf{H}_{-p}^{-1}\hat{\mathbf{X}}\mathbf{r}^\top + \mathbf{r}\mathbf{r}^\top. \quad (20)$$

An important observation is that $\mathbf{e}_p\mathbf{H}_{-p}^{-1}$ is in fact an all-zero vector since the $p$-th row of $\mathbf{H}_{-p}^{-1}$ is eliminated. Consequently, the third term in Equation 20 equals zero. Part of the second term can also be further simplified: $\mathbf{H}^{-1}\mathbf{e}_p^\top\mathbf{e}_p\mathbf{H}_{:,p}^{-1} = \mathbf{H}_{:,p}^{-1}\mathbf{H}_{pp}^{-1}$. The expression $\mathbf{H}_{p,:}^{-1}\mathbf{H}\mathbf{H}_{:,p}^{-1}$ in the first term can also be simplified to $\mathbf{H}_{pp}^{-1}$. Combining all these simplifications, Equation 20 reduces to:

$$\mathcal{L} = \frac{\mathbf{w}_p^2}{\mathbf{H}_{pp}^{-1}} + 2\frac{\mathbf{w}_p}{\mathbf{H}_{pp}^{-1}}\mathbf{r}\hat{\mathbf{X}}^\top\mathbf{H}_{:,p}^{-1} - \mathbf{r}\hat{\mathbf{X}}^\top\mathbf{H}_{-p}^{-1}\hat{\mathbf{X}}\mathbf{r}^\top + \mathbf{r}\mathbf{r}^\top. \quad (21)$$

By reordering the second and fourth terms, we obtain the identical format of $\mathcal{L}$ as presented in Equation 3.

## A.2 Proof of Cholesky Decomposition of $\mathbf{H}_{-p}^{-1}$ (Efficient calculation of $\mathbf{b}$ for pruning score $\mathbf{S}$)

Recall that in Section 4.3, we leverage the equality $\mathbf{H}_{-p}^{-1} = \mathbf{L}_{p+1:,p+1:}\mathbf{L}_{p+1:,p+1:}^\top$ to simplify the computation of term $\mathbf{b}$ for pruning score $\mathbf{S}$. We provide the proof for this equality using mathematical induction below.

*Proof.* We begin the mathematical induction proof by establishing the base case: $p = 1$.

**Base Case:** First, we rewrite the lower-triangular Cholesky factor $\mathbf{L}$ as:

$$\mathbf{L} = \begin{pmatrix} \mathbf{L}_{11} & \mathbf{0} \\ \mathbf{L}_{2:,1} & \mathbf{L}_{2:,2:} \end{pmatrix}. \quad (22)$$

The Cholesky decomposition of $\mathbf{H}$ can be written as:

$$\mathbf{H}^{-1} = \begin{pmatrix} \mathbf{H}_{11}^{-1} & \mathbf{H}_{1,2:}^{-1} \\ \mathbf{H}_{2:,1}^{-1} & \mathbf{H}_{2:,2:}^{-1} \end{pmatrix} = \begin{pmatrix} \mathbf{L}_{11} & \mathbf{0} \\ \mathbf{L}_{2:,1} & \mathbf{L}_{2:,2:} \end{pmatrix}\begin{pmatrix} \mathbf{L}_{11} & \mathbf{L}_{2:,1}^\top \\ \mathbf{0} & \mathbf{L}_{2:,2:}^\top \end{pmatrix}. \quad (23)$$

Based on this equation, we can construct the following linear system:

$$
\begin{cases}
\mathbf{H}_{11}^{-1} = \mathbf{L}_{11}^2 \\
\mathbf{H}_{2:,1}^{-1} = \mathbf{L}_{11}\mathbf{L}_{2:,1} \\
\mathbf{H}_{2:,2:}^{-1} = \mathbf{L}_{2:,1}\mathbf{L}_{2:,1}^\top + \mathbf{L}_{2:,2:}\mathbf{L}_{2:,2:}^\top
\end{cases}
. \tag{24}
$$

Solving the above equations, we obtain the expression for $\mathbf{L}_{2:,2:}\mathbf{L}_{2:,2:}^\top$ as:

$$
\begin{aligned}
\mathbf{L}_{2:,2:}\mathbf{L}_{2:,2:}^\top &= \mathbf{H}_{2:,2:}^{-1} - \frac{1}{\sqrt{\mathbf{H}_{11}^{-1}}}\mathbf{H}_{2:,1}^{-1}\frac{1}{\sqrt{\mathbf{H}_{11}^{-1}}}(\mathbf{H}_{2:,1}^{-1})^\top \\
&= \mathbf{H}_{2:,2:}^{-1} - \frac{1}{\mathbf{H}_{11}^{-1}}\mathbf{H}_{2:,1}^{-1}\mathbf{H}_{1,2:}^{-1}.
\end{aligned}
\tag{25}
$$

Recalling the expression for Gaussian elimination introduced in Section 3, Equation 25 essentially represents the removal of the first row and column from $\mathbf{H}^{-1}$ after applying Gaussian elimination to zero out its first row and column. Therefore, we have proven the base case that $\mathbf{H}_{-1}^{-1} = \mathbf{L}_{2:,2:}\mathbf{L}_{2:,2:}^\top$.

**Inductive Step:** Assume the statement holds for $p = k - 1$. We show it also holds for $p = k$. Since we know $\mathbf{H}_{-(k-1)}^{-1} = \mathbf{L}_{k:,k:}\mathbf{L}_{k:,k:}^\top$, the lower-triangular Cholesky factor $\mathbf{L}$ for $\mathbf{H}_{-(k-1)}^{-1}$ can be formulated as:

$$
\mathbf{L} = \begin{pmatrix} \mathbf{L}_{kk} & \mathbf{0} \\ \mathbf{L}_{(k+1):,k} & \mathbf{L}_{(k+1):,(k+1):} \end{pmatrix}. \tag{26}
$$

This formulation is valid because, under our inductive hypothesis for $p = k - 1$, all rows and columns before $k$ have already been removed. We can construct similar linear systems as in the base case:

$$
\begin{cases}
\mathbf{H}_{kk}^{-1} = \mathbf{L}_{kk}^2 \\
\mathbf{H}_{(k+1):,k}^{-1} = \mathbf{L}_{kk}\mathbf{L}_{(k+1):,k} \\
\mathbf{H}_{(k+1):,(k+1):}^{-1} = \mathbf{L}_{(k+1):,k}\mathbf{L}_{(k+1):,k}^\top + \mathbf{L}_{(k+1):,(k+1):}\mathbf{L}_{(k+1):,(k+1):}^\top
\end{cases}
. \tag{27}
$$

Solving the above equations, we again obtain:

$$
\begin{aligned}
\mathbf{L}_{(k+1):,(k+1):}\mathbf{L}_{(k+1):,(k+1):}^\top &= \mathbf{H}_{(k+1):,(k+1):}^{-1} - \frac{1}{\sqrt{\mathbf{H}_{kk}^{-1}}}\mathbf{H}_{(k+1):,k}^{-1}\frac{1}{\sqrt{\mathbf{H}_{kk}^{-1}}}(\mathbf{H}_{(k+1):,k}^{-1})^\top \\
&= \mathbf{H}_{(k+1):,(k+1):}^{-1} - \frac{1}{\mathbf{H}_{kk}^{-1}}\mathbf{H}_{(k+1):,k}^{-1}\mathbf{H}_{k,(k+1):}^{-1}.
\end{aligned}
\tag{28}
$$

Again, Equation 28 is equivalent to removing the first row and column of $\mathbf{H}_{-(k-1)}^{-1}$ after applying Gaussian Elimination to zero out its first row and column. This operation is equivalent to computing $\mathbf{H}_{-k}^{-1}$. Therefore, we have proven that $\mathbf{H}_{-k}^{-1} = \mathbf{L}_{(k+1):,(k+1):}\mathbf{L}_{(k+1):,(k+1):}^\top$. By the principle of mathematical induction, the statement holds for all $p$ with $k \geq 1$.

### A.3 Proof of $\mathbf{H}_{:,p}^{-1}/\mathbf{H}_{pp}^{-1} = \mathbf{L}_{:,p}/\mathbf{L}_{pp}$ (Efficient calculation of c for pruning score S)

This equality has previously been leveraged to simplify the implementation of SparseGPT (Frantar & Alistarh 2023) and GPTQ (Frantar et al. 2022). We also leverage this relationship to simplify the calculation of term **c** in our pruning score **S**.

*Proof.* We assume that at the $p$-th iteration of the calibration, all columns and rows before $p$ have been removed through Gaussian elimination. Based on this assumption, the Cholesky decomposition of the Hessian inverse at the $p$-th iteration can be formulated as:

$$
\mathbf{H}_{-(p-1)}^{-1} = \begin{pmatrix} \mathbf{H}_{pp}^{-1} & \mathbf{H}_{p,(p+1):}^{-1} \\ \mathbf{H}_{(p+1):,p}^{-1} & \mathbf{H}_{(p+1):,(p+1):}^{-1} \end{pmatrix} = \begin{pmatrix} \mathbf{L}_{pp} & \mathbf{0} \\ \mathbf{L}_{(p+1):,p} & \mathbf{L}_{(p+1):,(p+1):} \end{pmatrix} \begin{pmatrix} \mathbf{L}_{pp} & \mathbf{L}_{(p+1):,p}^\top \\ \mathbf{0} & \mathbf{L}_{(p+1):,(p+1):}^\top \end{pmatrix}.
\tag{29}
$$

This equation also leverages the result proven in Section A.2. Next, we divide the expression $\mathbf{H}_{:,p}^{-1}$ into three parts: $\mathbf{H}_{1:(p-1),p}^{-1}$, $\mathbf{H}_{pp}^{-1}$, and $\mathbf{H}_{(p+1):,p}^{-1}$. For the first two parts, the equality is straightforward

to prove. Since $\mathbf{H}_{1:(p-1),p}^{-1} = \mathbf{0}^\top = \mathbf{L}_{1:(p-1),p}$, so $\mathbf{H}_{1:(p-1),p}^{-1}/\mathbf{H}_{pp}^{-1} = \mathbf{0}^\top = \mathbf{L}_{1:(p-1),p}/\mathbf{L}_{pp}$. And $\mathbf{H}_{pp}^{-1}/\mathbf{H}_{pp}^{-1} = 1 = \mathbf{L}_{pp}/\mathbf{L}_{pp}$. The only equality that requires proof is: $\mathbf{H}_{(p+1):,p}^{-1}/\mathbf{H}_{pp}^{-1} = \mathbf{L}_{(p+1):,p}/\mathbf{L}_{pp}$. From Equation 29, we can construct the following linear relations:

$$\begin{cases} \mathbf{H}_{pp}^{-1} = \mathbf{L}_{pp}^2 \\ \mathbf{H}_{(p+1):,p}^{-1} = \mathbf{L}_{pp}\mathbf{L}_{(p+1):,p} \end{cases}. \tag{30}$$

Substituting these relations back into the expression, it is straightforward to verify that $\mathbf{H}_{(p+1):,p}^{-1}/\mathbf{H}_{pp}^{-1} = \mathbf{L}_{pp}\mathbf{L}_{(p+1):,p}/\mathbf{L}_{pp}^2 = \mathbf{L}_{(p+1):,p}/\mathbf{L}_{pp}$.

## A.4 Proof of $\mathbf{U} = \left((\Delta\mathbf{X}\hat{\mathbf{X}}^\top\mathbf{L}) \odot \mathbf{M}^\mathbf{u}\right)$

Recall that in Section 4.2, $\mathbf{b}$ is first proposed to be precomputed as $\mathbf{b} = \mathrm{diag}(\mathbf{U}\mathbf{U}^\top)$, where $\mathbf{U}_{p,:} = \Delta\mathbf{X}_{p,:}\hat{\mathbf{X}}^\top\mathbf{L}_{p+1:,p+1:}$. To further improve the algorithmic efficiency, we propose to re-express $\mathbf{U}$ in the form $\left((\Delta\mathbf{X}\hat{\mathbf{X}}^\top\mathbf{L}) \odot \mathbf{M}^\mathbf{u}\right)$, so that the term $\Delta\mathbf{X}\hat{\mathbf{X}}^\top\mathbf{L}$ can be reused between $\mathbf{b}$ and $\mathbf{c}$. The proof is straightforward and provided below.

*Proof.* The $p$-th row of $\mathbf{U}$ equals $\Delta\mathbf{X}_{p,:}\hat{\mathbf{X}}^\top\mathbf{L}_{p+1:,p+1:}$, where $\mathbf{L} \in \mathcal{R}^{k \times k}$ is the lower-triangle Cholesky factor. For any given vector $\mathbf{z} \in \mathcal{R}^{1 \times k}$ and the Cholesky lower-triangle factor matrix $\mathbf{L}_{p+1:,p+1:}$, we have:

$$(\mathbf{z}\mathbf{L}_{p+1:,p+1:})_i = \begin{cases} 0 & \text{if } i < p+1 \\ \mathbf{z}\mathbf{L}_{:,i} & \text{else } i \geq p+1 \end{cases}. \tag{31}$$

Therefore, substituting $\mathbf{z} = \Delta\mathbf{X}_{p,:}\hat{\mathbf{X}}^\top \in \mathcal{R}^{1 \times k}$ into the above equation, we can reformulate the $p$-th row of $\mathbf{U}$ as:

$$\mathbf{U}_{p,i} = \begin{cases} 0 & \text{if } i < p+1 \\ \Delta\mathbf{X}_{p,:}\hat{\mathbf{X}}^\top\mathbf{L}_{:,i} & \text{else } i \geq p+1 \end{cases}. \tag{32}$$

Hence, $\mathbf{U}_{p,:}$ equals $(\Delta\mathbf{X}_{p,:}\hat{\mathbf{X}}^\top\mathbf{L}) \odot \mathbf{M}_{p,:}^\mathbf{u}$, where $\mathbf{M}^\mathbf{u}$ is a strictly upper-triangular masking matrix with ones above the diagonal. The full computation of $\mathbf{U}$ can thus be re-expressed as $\mathbf{U} = (\Delta\mathbf{X}\hat{\mathbf{X}}^\top\mathbf{L}) \odot \mathbf{M}^\mathbf{u}$. By leveraging the independence across rows of $\Delta\mathbf{X}$, $\mathbf{b} = \mathrm{diag}(\mathbf{U}\mathbf{U}^\top)$, where $\mathbf{U} = (\Delta\mathbf{X}\hat{\mathbf{X}}^\top\mathbf{L}) \odot \mathbf{M}^\mathbf{u}$.

## A.5 Proof of Theroem 1

The key step in proving Equation 8 is to solve for $\mathcal{L}_{\mathrm{SparseGPT}}$. We will follow the same procedure as in Section A.1. According to Equation 1, the optimal weight update for SparseGPT is $\Delta\mathbf{w} = -\frac{\mathbf{w}_p}{\mathbf{H}_{pp}^{-1}} \cdot \mathbf{H}_{p,:}^{-1}$. Substituting this into the loss function $\mathcal{L} = ||\Delta\mathbf{w}\hat{\mathbf{X}} - \mathbf{r}||_F^2$, the loss for SparseGPT becomes:

$$\mathcal{L}_{\mathrm{SparseGPT}} = \left|\left| -\frac{\mathbf{w}_p}{\mathbf{H}_{pp}^{-1}}\mathbf{H}_{p,:}^{-1}\hat{\mathbf{X}} - \mathbf{r} \right|\right|_F^2. \tag{33}$$

Given the loss of **DuoGPT** in Equation 17, it is straightforward to observe that the difference between the two loss functions after equation expansion is simply $\Delta\mathcal{L} = \mathbf{r}\hat{\mathbf{X}}^\top\mathbf{H}_{-p}^{-1}\hat{\mathbf{X}}\mathbf{r}^\top$.

The first observation we can make is that, since $\mathbf{H}_{-p}^{-1}$ is positive definite, the loss improvement of **DuoGPT** ($\Delta\mathcal{L}$) will always be greater than or equal to zero. Then, since $\mathbf{H}_{-p}^{-1}$ is obtained from $\mathbf{H}^{-1}$ via Gaussian elimination, it also follows the spectral bound that $\lambda_{\min}(\mathbf{H}_{-p}^{-1}) \geq \frac{\alpha}{\lambda_{\max}(\mathbf{H})}$ where $\alpha > 0$ is the stability constant. We can then rewrite $\Delta\mathcal{L}$ into its quadratic form to get $\Delta\mathcal{L} \geq \frac{\alpha||\mathbf{r}\hat{\mathbf{X}}^\top||^2}{\lambda_{\max}(\mathbf{H})}$. Here, it is reasonable to bound $||\mathbf{r}\hat{\mathbf{X}}^\top||^2$ as $||\mathbf{w}\sigma_r||^2$, where $\sigma_r$ is just a constant to capture the intensity of the activation residual. This approximation is reasonable due to the fact that we are always perturbing the smallest values during calibration, so the term is not large overall. After assuming a weight norm lower bound of $C_\mathbf{w}$, we can get $\Delta\mathcal{L} \geq \frac{\alpha C_\mathbf{w}^2 \sigma_r^2}{\lambda_{\max}(\mathbf{H})}$. Finally, it is intuitive that the activation residual intensity is proportional to the number of "zeroed-out" activations; thus, we include the term $\mathbf{p}^\mathbf{x}m$ to capture the context of global activation residual intensity. This yields the final error bound as shown in Equation 8.

# B  Detailed Experimental Setups

This section provides comprehensive implementation details and experimental configurations to ensure reproducibility of our results.

## B.1  Calibration Settings for DuoGPT and Unstructured Baselines

We implement all unstructured baselines reported in Table 1 using uniform evaluation settings to ensure fair comparison. Our implementation builds upon the calibration framework from SparseGPT (Frantar & Alistarh 2023), using PyTorch (Paszke 2019).

For both SparseGPT and **DuoGPT**, we enable *act_order*, an option in SparseGPT that sorts weight columns based on Hessian diagonal magnitude to improve pruning performance. The dampening ratio for the Hessian is set to 0.1 for numerical stability. We apply the lazy batch updates and iterative mask blocking techniques from SparseGPT to **DuoGPT** for reducing I/O overhead. The block size and lazy batch size are both set to $B = 128$. For both methods, we prune the input calibration data to each layer based on magnitude before calculating each layer's Hessians.

We integrate the original pruning and mask selection procedures from Wanda (Sun et al. 2023) into our evaluation framework. Since Wanda lacks a compensation mechanism post-pruning, we maintain dense activation throughout the calibration process (activation sparsity-aware pruning disabled).

For 2:4 structured versions of both SparseGPT and Wanda, we apply mask selection and pruning over every 4-column group with 50% weight sparsity. All other configurations remain identical to their unstructured counterparts. Calibration data remains dense throughout the pruning process for all 2:4 baselines.

## B.2  Settings for Speedup Results

We report end-to-end GPU speedup results on an NVIDIA A100 80GB GPU in Table 2 and 6. All speedups are normalized with respect to the dense model baseline rather than reporting absolute running times to ensure fair comparison across different frameworks.

For **DuoGPT**, end-to-end GPU speedup is measured by inheriting TEAL's custom Triton kernel (Liu et al. 2025) and integrating it into the fast-gpt framework. The corresponding dense model baseline is also evaluated using the fast-gpt framework to obtain normalized speedup ratios. All speedups for structured pruned models are measured using 2SSP's framework (Sandri et al. 2025), with dense model baselines similarly evaluated within the same framework to compute normalized speedup.

## B.3  Settings to Measure the Number of Weights Loaded to SRAM and Model Size

In Figure 3, we report the number of weights loaded into SRAM during inference. For dense models, structured-sparse models, and activation-sparse models, the number of weights loaded into SRAM is fixed. However, for **DuoGPT**, the dual-sparsity approach combined with unstructured weight distribution across rows makes the number of loaded weights a dynamic quantity that depends on which activation patterns are selected. To ensure a conservative and fair comparison, we report the worst-case SRAM loading numbers for **DuoGPT**. Specifically, we sort weight rows by their weight sparsity and assume that dynamic activation sparsity always selects the rows with the lowest weight sparsity (i.e., the densest rows requiring maximum SRAM access). For example, with 40% activation sparsity and 40% unstructured weight sparsity, we first sort all weight rows by their sparsity levels, then select the 40% of rows with the lowest weight sparsity—those requiring the largest number of weights to be loaded into SRAM. This conservative methodology ensures our reported SRAM access numbers represent an upper bound; in practice, **DuoGPT**'s savings on SRAM accesses can be even higher.

Model size results reported in Table 3(b) and Table 5 include the full parameter count, including all model parameters beyond the transformer blocks (e.g., embedding tables and output projection layers).

# C   Extra Experimental Results

## C.1   Extra Performance Comparisons with Structured Pruning Baselines

In Table 6, we provide additional comparison results against structured pruning baselines on LLaMA-2-13B. All structured pruning baselines are pruned with 30% weight-only sparsity. The structured pruning methods use 256 calibration samples (except 2SSP, which uses 32 samples following its original setup) with 2048 token sequences. We employ 50% dual-sparsity for **DuoGPT**. The results demonstrate that **DuoGPT** achieves better performance across speedup, model size, and accuracy metrics. Notably, 50% dual-sparse **DuoGPT** achieves even higher speedup (1.51×) compared to the 30% structured pruned models, where the fastest baseline achieves 1.41× speedup.

Table 6: Extra comparison result with other structured pruning methods on LLaMA-2-13B.

| Model | Method | Model Size | Speedup | Wiki2($\downarrow$) | PIQA | HellaSw | ARC-E | ARC-C | WinoG | Avg($\uparrow$) |
|---|---|---|---|---|---|---|---|---|---|---|
| LLaMA-2-13B | ShortGPT | 9.21B/13.02B | 1.41× | 39.78 | 69.80 | 57.94 | 52.86 | 35.75 | **69.06** | 57.08 |
| | 2SSP | 9.21B/13.02B | 1.29× | 9.07 | **76.66** | **70.51** | 65.28 | 38.74 | 68.90 | 64.02 |
| | BlockPruner | 9.21B/13.02B | 1.38× | 9.67 | 73.94 | 64.39 | 61.87 | 37.37 | 66.61 | 60.84 |
| | **DuoGPT** | **6.67B**/13.02B | **1.51×** | **7.17** | 75.63 | 69.56 | **69.95** | **41.13** | 68.11 | **64.88** |

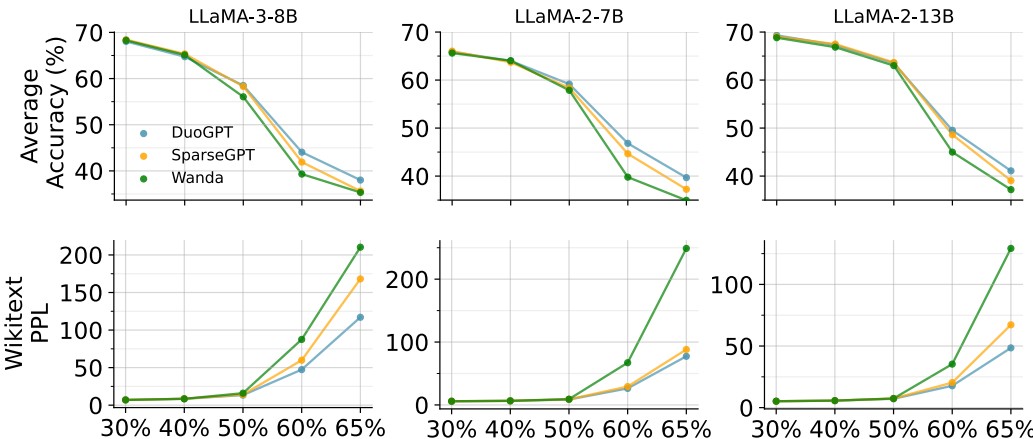

Figure 4: Mean zero-shot accuracy and perplexity on LLaMA-3-8B, LLaMA-2-7B, and LLaMA-2-13B. The results are reported across different dual-sparsity levels. The perplexity is reported for WikiText2 dataset and the accuracy results are averaged across 7 tasks.

## C.2   Performance across Different Dual-Sparsity Levels

We provide a comprehensive visualization of our approach's performance compared to baselines across different dual-sparsity levels in Figure 4. The two baselines are dual-sparse variants of SparseGPT and Wanda, using identical setups as those in Table 1. We plot perplexity (PPL) performance on the WikiText2 dataset and mean zero-shot accuracy across 7 tasks. We visualize sparsity levels of 30%, 40%, 50%, 60%, and 65%. Beyond 65% dual-sparsity, performance degrades too significantly from the dense model to be meaningful, while below 30%, efficiency gains are insufficient to justify the complexity.

Several key trends emerge from Figure 4. At low dual-sparsity regimes (<50%), performance differences between methods are minimal regardless of calibration settings. However, the benefits of activation sparsity-aware calibration become pronounced at high dual-sparsity regimes ($\geq$50%). For example, Wanda's perplexity performance (without activation sparsity-aware calibration) begins to degrade exponentially beyond 60% dual-sparsity. The SparseGPT baseline partially recovers this performance loss through activation sparsity-aware calibration, though these benefits diminish at even higher sparsity levels (65%).

**DuoGPT** addresses this limitation through output residual compensation, using dense model outputs to compensate for information loss during sparse calibration. This enables superior performance

in extreme dual-sparsity regimes, such as 65%. This trend is consistently observed across different metrics (perplexity and zero-shot accuracy) and model scales, as visualized in Figure 4. Detailed perplexity and zero-shot accuracy results for each task are provided in Table 7 for reference.

Table 7: Detailed zero-shot accuracy results for LLaMA-3-8B, LLaMA-2-7B, and LLaMA-2-13B across different dual-sparsity levels. ARC-E stands for ARC-Easy, ARC-C stands for ARC-Challenge, WinoG stands for WinoGrande, and OBQA stands for OpenBookQA. Bold and underlined values indicate the best and second-best results, respectively.

| Model | Method | Wiki2(↓) | PIQA | HellaSwag | ARC-E | ARC-C | WinoG | BoolQ | OBQA | Avg(↑) |
|---|---|---|---|---|---|---|---|---|---|---|
| **LLaMA-3-8B** $W_{30\%}X_{30\%}$ | SparseGPT | 7.00 | 78.94 | **77.20** | **76.43** | 49.06 | **72.85** | **81.07** | 43.40 | **68.42** |
| | Wanda | 6.97 | **79.27** | 77.10 | 75.63 | **50.17** | 71.82 | 80.18 | 43.80 | 68.28 |
| | **DuoGPT** | **6.96** | 78.35 | 77.08 | 75.84 | 48.55 | 72.14 | 80.92 | 43.40 | 68.04 |
| $W_{40\%}X_{40\%}$ | SparseGPT | 8.57 | **77.26** | 73.07 | **71.68** | 44.97 | **70.88** | **78.81** | 40.80 | **65.35** |
| | Wanda | **8.46** | 76.28 | 72.61 | 71.42 | **46.59** | 69.22 | 77.58 | **42.20** | 65.13 |
| | **DuoGPT** | 8.47 | 76.88 | **73.21** | 70.88 | 44.28 | 68.11 | 78.59 | 41.20 | 64.74 |
| $W_{50\%}X_{50\%}$ | SparseGPT | 14.05 | 72.91 | **61.96** | 62.29 | **37.71** | 62.43 | 74.62 | 36.20 | 58.30 |
| | Wanda | 15.98 | 71.44 | 57.01 | 59.34 | 35.84 | 61.33 | 70.80 | **36.60** | 56.05 |
| | **DuoGPT** | **13.41** | **73.72** | 61.88 | **62.96** | 36.77 | **64.72** | 74.25 | 35.20 | **58.50** |
| $W_{60\%}X_{60\%}$ | SparseGPT | 59.85 | 58.32 | 33.96 | 39.39 | 23.38 | 52.25 | 58.29 | **27.80** | 41.91 |
| | Wanda | 87.50 | 56.64 | 30.64 | 35.02 | 20.82 | 50.99 | 55.29 | 25.80 | 39.31 |
| | **DuoGPT** | **47.33** | **61.32** | **36.95** | **42.76** | **24.83** | **52.57** | **62.29** | 27.60 | **44.05** |
| $W_{65\%}X_{65\%}$ | SparseGPT | 59.85 | 54.35 | 28.22 | 29.76 | 21.33 | 50.99 | 38.56 | 26.20 | 35.63 |
| | Wanda | 87.50 | 53.37 | 28.18 | 28.79 | **22.44** | 50.51 | 38.04 | 26.00 | 35.33 |
| | **DuoGPT** | **47.33** | **56.75** | **28.91** | **32.74** | 20.56 | **51.46** | **49.48** | **26.20** | **38.01** |
| **LLaMA-2-7B** $W_{30\%}X_{30\%}$ | SparseGPT | 5.87 | **78.45** | 74.97 | **73.57** | 45.39 | **69.14** | 77.06 | 43.60 | **66.03** |
| | Wanda | 5.85 | 78.29 | **75.30** | 73.32 | **46.16** | 67.56 | 76.30 | 42.40 | 65.62 |
| | **DuoGPT** | **5.84** | 77.91 | 74.66 | 73.06 | 45.65 | 68.43 | **77.34** | **44.00** | 65.86 |
| $W_{40\%}X_{40\%}$ | SparseGPT | 6.52 | 77.04 | 71.80 | 70.12 | 42.75 | 66.30 | 76.12 | 41.80 | 63.70 |
| | Wanda | 6.50 | **77.26** | **72.01** | 69.99 | 42.92 | **67.64** | 75.60 | **42.80** | **64.03** |
| | **DuoGPT** | **6.48** | 76.61 | 71.57 | **71.13** | **43.34** | 67.17 | **76.18** | 42.00 | 64.00 |
| $W_{50\%}X_{50\%}$ | SparseGPT | 8.98 | **74.43** | **63.89** | 62.67 | **35.75** | 63.46 | 71.56 | 36.80 | 58.37 |
| | Wanda | 9.13 | 73.01 | 62.30 | 61.66 | 35.32 | 62.75 | 70.24 | **39.80** | 57.87 |
| | **DuoGPT** | **8.58** | 73.83 | 63.74 | **64.18** | 35.67 | **65.35** | **72.54** | 38.80 | **59.16** |
| $W_{60\%}X_{60\%}$ | SparseGPT | 29.21 | 61.15 | 38.20 | 42.38 | 24.57 | 52.41 | **63.67** | 30.20 | 44.65 |
| | Wanda | 67.13 | 55.93 | 29.67 | 34.01 | 22.70 | 50.59 | 58.59 | 27.00 | 39.78 |
| | **DuoGPT** | **26.27** | **63.49** | **41.54** | **45.03** | **26.96** | **56.59** | 63.30 | **30.80** | **46.82** |
| $W_{65\%}X_{65\%}$ | SparseGPT | 88.17 | 53.43 | 29.04 | 29.25 | 22.53 | 49.49 | 51.38 | 25.60 | 37.25 |
| | Wanda | 248.9 | 52.18 | 28.10 | 28.16 | **24.32** | 50.20 | 38.26 | 23.40 | 34.95 |
| | **DuoGPT** | **77.34** | **55.44** | **29.80** | **32.62** | 21.59 | **50.83** | **58.90** | **28.60** | **39.68** |
| **LLaMA-2-13B** $W_{30\%}X_{30\%}$ | SparseGPT | 5.20 | 79.71 | **78.62** | 75.88 | **49.66** | 71.90 | 80.83 | 46.80 | 69.06 |
| | Wanda | 5.22 | 79.43 | 78.56 | **75.80** | 49.49 | 71.35 | 80.12 | 47.00 | 68.82 |
| | **DuoGPT** | **5.18** | **80.36** | 78.57 | 75.46 | 49.15 | 71.74 | **81.77** | **48.20** | **69.32** |
| $W_{40\%}X_{40\%}$ | SparseGPT | 5.70 | **79.33** | 76.45 | **73.36** | **47.10** | **71.98** | **81.10** | 43.20 | **67.50** |
| | Wanda | 5.69 | 78.56 | **76.73** | 72.43 | 46.42 | 68.51 | 79.88 | **45.40** | 66.85 |
| | **DuoGPT** | **5.66** | 78.67 | 75.97 | 73.02 | 47.01 | 70.80 | 80.83 | 44.80 | 67.30 |
| $W_{50\%}X_{50\%}$ | SparseGPT | 7.39 | 76.28 | 69.14 | 68.52 | **41.81** | **68.67** | 79.30 | 42.00 | **63.67** |
| | Wanda | 7.41 | **77.20** | 69.15 | 67.93 | 40.19 | 66.54 | 77.28 | 42.80 | 63.01 |
| | **DuoGPT** | **7.17** | 76.12 | **69.43** | **69.49** | 40.96 | 67.40 | **77.83** | **43.40** | 63.52 |
| $W_{60\%}X_{60\%}$ | SparseGPT | 20.33 | 64.74 | 43.20 | 49.83 | 28.33 | 56.27 | **68.13** | 29.80 | 48.61 |
| | Wanda | 35.33 | 63.00 | 36.90 | 44.65 | 25.09 | 52.17 | 62.72 | 30.40 | 44.99 |
| | **DuoGPT** | **17.74** | **66.38** | **45.74** | **50.29** | **28.41** | **57.06** | 67.25 | **31.60** | **49.53** |
| $W_{65\%}X_{65\%}$ | SparseGPT | 67.21 | 54.90 | 29.76 | 30.77 | 20.90 | 49.41 | 61.87 | 25.80 | 39.06 |
| | Wanda | 129.22 | 52.50 | 29.31 | 29.04 | **22.44** | 51.07 | 50.70 | 25.20 | 37.18 |
| | **DuoGPT** | **48.49** | **58.16** | **31.76** | **34.34** | 22.35 | **51.62** | **62.20** | **27.20** | **41.09** |

## C.3   Sensitivity Analysis on the Calibration Set Size and Sequence Length

We present an ablation study to analyze the role of the C4 calibration dataset. We focus on **DuoGPT** with 50% dual-sparsity using the LLaMA-2-7B model. For the calibration set size study, we fix the sequence length to 2048 tokens. For the calibration sequence length study, we fix the number of calibration samples to 128.

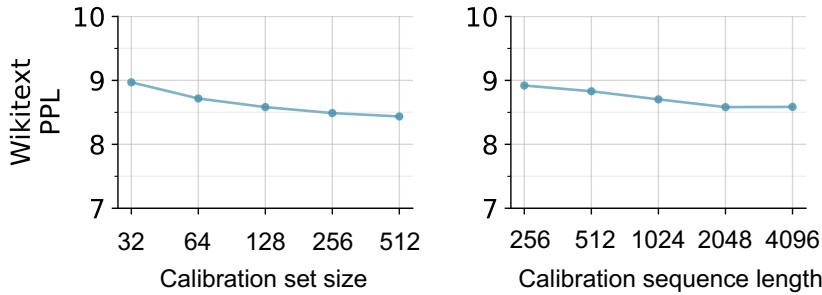

Figure 5: The effect of the C4 calibration set size and sequence length on PPL of WikiText2 dataset for LLaMA-2-7B.

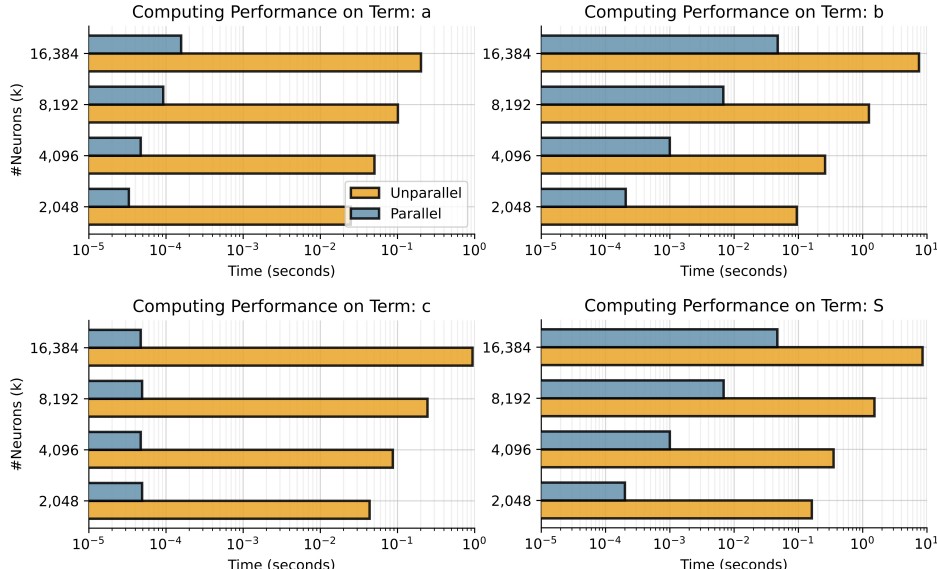

Figure 6: Latency visualization of our algorithm under different size of input channel $k$.

Figure 5 (left) shows the effect of varying the number of calibration samples on WikiText2 perplexity. The results demonstrate that at least 128 calibration samples provide reasonable performance for our calibration procedure.

We next explore the effect of different sequence lengths in the calibration dataset. Interestingly, we find that beyond a sequence length of 2048 tokens, the perplexity does not continue to decrease, as shown in Figure 5 (right). We conclude that for activation sparsity-aware calibration, a sequence length of 2048 tokens is sufficient to achieve good perplexity performance.

### C.4 Algorithm Efficiency

Finally, we provide a visualization of the GPU runtime for our efficient **DuoGPT** implementation (Equation 7) compared to the unparallelized implementation (Equation 6). We measure latency on a single A100 80GB GPU using PyTorch 2.4.0. We provide 10 warm up runs. The token dimension ($m$) is fixed at 2048. Figure 6 compares the latency for computing $\mathbf{a}$, $\mathbf{b}$, $\mathbf{c}$, and the overall pruning score $\mathbf{S}$. Owing to highly optimized CUDA kernels, our vectorized precomputation of all quantities completes in approximately 1ms for a $4096 \times 4096$ layer with $m = 2048$ tokens, achieving roughly $350\times$ speedup over the unparallelized implementation. Note that we do not compare against the naive implementation that follows the optimal pruning order, as it is prohibitively slow and would not yield meaningful comparisons.

## C.5 Extra Comparison with Prior Works

Given that CATS (Lee et al. 2024a) is an important prior work that validated the use of a thresholding technique to induce activation sparsity in LLMs, it is important to compare **DuoGPT** against it. We set the comparison to be at 40% global sparsity, which means that **DuoGPT** will have 40% dual-sparsity. Since CATS only explores activation sparsity in FFN layers, it requires 89.7% activation sparsity in those layers to achieve an equivalent 40% global sparsity. The comparison between the two works on 5 downstream tasks is reported in Table 8. Since we directly report the results from CATS, for a fair comparison, we do not report the normalized accuracy for **DuoGPT** in this comparison.

Table 8: Comparison with CATS on LLaMA-2-7B at 40% global sparsity.

| Method | PIQA | HellaSw | ARC-E | ARC-C | WinoG | Avg($\uparrow$) |
|---|---|---|---|---|---|---|
| CATS | 66.27 | 38.48 | 45.66 | 28.16 | 57.38 | 47.19 |
| **DuoGPT** | **76.66** | **53.42** | **74.37** | **40.78** | **67.17** | **62.48** |

For reference, we also provide an additional comparison with ReLUfication (Mirzadeh et al. 2023). Since ReLU mostly yields around 50% activation sparsity, we compare it with **DuoGPT** at 22% global dual-sparsity. As discussed in (Lee et al. 2024a), ReLU-based methods perform poorly without enough epochs of fine-tuning.

Table 9: Comparison with ReLUfication on LLaMA-2-7B at 22% global sparsity.

| Method | PIQA | HellaSw | ARC-E | ARC-C | WinoG | Avg($\uparrow$) |
|---|---|---|---|---|---|---|
| ReLUfication | 54.08 | 25.86 | 27.95 | 24.06 | 48.93 | 36.18 |
| **DuoGPT** | **78.73** | **56.80** | **76.47** | **43.26** | **67.96** | **64.64** |

## C.6 Isolated Effect of Asymmetric Calibration on Weight-only Pruning

It is useful to provide the isolated effect of asymmetric calibration (Li et al. 2025) on weight-only pruning, so that we can get a better idea of how activation sparsity-aware asymmetric calibration helps improve performance. We evaluate two scenarios: (1) weight-only pruning (no activation sparsity), and (2) dual-sparsity (with activation sparsity). We report the WikiText2 PPL results at 50% sparsity across different models in Table 10.

Table 10: Ablation study isolating residual correction effects.

| Scenario | Method | LLaMA-2-7B | LLaMA-2-13B | LLaMA-3-8B |
|---|---|---|---|---|
| Weight-only | SparseGPT | 7.11 | 6.15 | 9.98 |
| | **DuoGPT** | **6.97** | **6.03** | **9.73** |
| Dual-sparsity | SparseGPT | 8.98 | 7.39 | 14.05 |
| | **DuoGPT** | **8.58** | **7.17** | **13.41** |

The results demonstrate the following: 1. Asymmetric correction is effective on its own: DuoGPT outperforms SparseGPT even in the weight-only pruning setting, showing that the second term in $\Delta\mathbf{w}$ (Equation 3) provides a clear benefit. 2. Activation sparsity amplifies the gain of asymmetric calibration. When activation sparsity is introduced, the magnitude of improvement increases by 2.9×, 1.8×, and 2.6×, respectively, across the three models. These results confirm that our closed-form solution adapts effectively: the asymmetric calibration compensates for error from both weight pruning and activation sparsity.

## C.7 Extra Results on Different LLM Architectures.

Demonstrating our method's generalizability across LLM model architectures is crucial. We have thus evaluated **DuoGPT** on multiple model families beyond LLaMA, which shows consistent

improvements across different architectures. We tested **DuoGPT** on Mistral (Albert et al. 2024), Qwen2 (Bai et al. 2023), and OPT (Zhang et al. 2022) of varying model sizes. We report the WikiText2 PPL at 50% dual-sparsity in Table 11.

Table 11: WikiText2 perplexity results across different LLM model families.

| Method | Mist-7B | Qwen-2-7B | Qwen-2-1.5B | OPT-125M | OPT-1.3B |
|---|---|---|---|---|---|
| Dense | 5.25 | 7.13 | 9.54 | 27.65 | 14.62 |
| SparseGPT | 8.40 | 10.91 | 19.50 | 55.08 | 35.25 |
| **DuoGPT** | 7.91 | 10.76 | 18.11 | 51.88 | 31.70 |

The results show that our method consistently outperforms SparseGPT-based dual-sparse baselines, demonstrating **DuoGPT**'s generalizability across different LLM model architectures.

