# OpenReview forum: "DuoGPT: Training-free Dual Sparsity through Activation-aware Pruning in LLMs"
_NeurIPS.cc/2025/Conference — NeurIPS 2025 poster_

### Official Review · Reviewer_MRdQ · 2025-06-02

**Clarity:** 3
**Significance:** 3
**Originality:** 3
**Rating:** 4
**Confidence:** 3

**Summary:**

This paper introduces DuoGPT, a training-free pruning framework for large language models (LLMs) that combines unstructured weight pruning with activation sparsity to create dual-sparse (spMspV) workloads. By reinterpreting activation sparsity as a dynamic, structured form of weight sparsity, the authors enhance the Optimal Brain Compression (OBC) framework with activation-aware calibration and dense-model residual correction. The framework is computationally efficient, scalable to billion-parameter models like LLaMA-3-70B, and calibrated within ~2 hours on a single A100 GPU.

**Questions:**

Thanks for this submission. Could you please address the following concerns:
1. Since Dual spasity has a closed-form and the accuracy is usually acceptable, I wonder if there is any more concise bound of error in terms of numeric values or higher-level semantic information. I mean, something like Section 3 in this paper: https://proceedings.neurips.cc/paper_files/paper/2016/file/0e55666a4ad822e0e34299df3591d979-Paper.pdf
2. I feel like OBC is not squeezing bits; therefore, is it orthogonal to symbol quantisations like INT4/INT8? If so, what if they are placed together?

**Ethical Concerns:**

["NO or VERY MINOR ethics concerns only"]

**Final Justification:**

My concerns are well-addressed, and I'd like to keep the score.

**Limitations:**

N.A. now

**Paper Formatting Concerns:**

N.A.

**Quality:**

3

**Strengths And Weaknesses:**

Strengthens:
1. Novel integration of activation sparsity and unstructured pruning.

2. Efficient closed-form solution with GPU-optimized implementation.

Weakness:
1. There is no discussion about the error bound.
2. Other LLM arches may be different from LLAMA, and theoretical clarifications or more evaluations may be required

---

> ### Author Rebuttal · Authors · 2025-07-31
>
> #### We thank the reviewer for the insightful questions and suggestions on error bound, quantization compatibility, and different LLM architectures. We have tried our best to address all of them in our rebuttal.
>
> > **Q1: Since dual sparsity has a closed-form and the accuracy is usually acceptable, I wonder if there is any more concise bound of error in terms of numeric values or higher-level semantic information.**
>
> #### Thank you for this important question and the reference. We provide a concise improvement bound quantifying DuoGPT's guaranteed loss improvement over SparseGPT below.
>
> #### **Theorem (DuoGPT Improvement Bound):** Under activation sparsity level $p_x$, DuoGPT achieves guaranteed loss improvement over SparseGPT:
>
> #### ${\Delta}L = L_{\text{SparseGPT}} - L_{\text{DuoGPT}} = r \hat{X}^T H^{-1}_{-p} \hat{X} r^T$.
>
> #### Then, the loss improvement satisfies the lower bound:
> #### $\geq \dfrac{\alpha p_x \sigma_r^2 C_W^2 m}{\lambda_{\max}(H)}$.
>
> #### where $\alpha > 0$ is the stability constant measuring spectral gap preservation, $\lambda_{\max}(H)$ is the maximum eigenvalue of the Hessian $H = \hat{X} \hat{X}^T$, $\sigma_r > 0$ is the activation residual magnitude constant, $C_W$ is the weight norm bound that satisfies $\||W\||_F^2 \leq C_W$, and $m$ is the calibration batch size.
>
> #### **Proof Sketch:**
>
> #### The key insight is that the weight update rules of SparseGPT and DuoGPT differ only in the residual correction term (Equation 3). Therefore we have the loss of SparseGPT as
>
> #### $L_{\text{SparseGPT}} = || -\frac{w_p}{H^{-1}_{pp}}||^2_F$, and the loss of DuoGPT from the Equation (20) in the appendix.
>
> #### When expanding both loss functions and subtracting $L_{\text{DuoGPT}}$ from $L_{\text{SparseGPT}}$, all terms cancel except one quadratic form: $r \hat{X}^T H^{-1}_{-p} \hat{X} r^T$.
>
> #### Since $H^{-1}_{-p}$ is positive definite, this guarantees $\Delta L \geq 0$ always.
>
> #### For the lower bound, we use the fact that $H^{-1}_{-p}$ is obtained from $H^{-1}$ via Gaussian elimination, which follows the spectral bound that
>
> #### $\lambda_{\min}(H^{-1}_{-p})$ is greater or equal to
>
> #### $\frac{\alpha}{\lambda_{\max}(H)}$. After applying the quadratic form bound where
>
> #### $r \hat{X}^T H^{-1}_{-p} \hat{X} r^T$ is greater or equal to
>
> #### $\lambda_{\min}(H^{-1}_{-p}) \cdot \||r \hat{X}^T\||^2$. Using activation sparsity properties yields our result.
>
> #### **Empirical Validation:** Our bound predicts meaningful improvement that scales linearly with activation sparsity $p_x$. Experimentally, we observe consistent improvement over SparseGPT across all models and sparsity levels in Table 1 and Table 3a, confirming the bound's validity. The linear scaling with $p_x$ explains why higher dual-sparsity yields greater improvements.
>
> #### We will add discussion of the bound to the revised version of the paper. The complete formal proof will be included in Appendix of the revised manuscript. We will make sure to reference the provided paper as our theoretical backup.
>
> > **Q2: Compatible with weight quantization?**
>
> #### Thank you for this question! You are correct about the orthogonal relationship between quantization and OBC-based pruning - they operate on different aspects of model compression and can be combined effectively.
>
> #### We evaluated the combination of INT8/INT4 uniform weight (group size 128) quantization with 50% dual-sparse DuoGPT on LLaMA-2-7B and 13B (WikiText2 PPL).
>
> #### DuoGPT maintains its performance advantage over SparseGPT across all precision levels, demonstrating successful orthogonal combination. Notably, our method shows better robustness to INT4 quantization, suggesting that the activation-sparsity-aware calibration helps preserve model quality under aggressive compression.
>
> #### This multi-dimensional compression approach is very promising for practical edge deployment-combining INT4 precision with 50% dual-sparsity achieves roughly 6% model density for SRAM loading, a compression ratio difficult to attain through any single compression method alone.
>
> | $\small{\text{Model}}$         | $\small{\text{Method}}$     | $\small{\text{FP16}}$ | $\small{\text{INT8}}$ | $\small{\text{INT4}}$ |
> |-------------------------------|------------------------------|------------------------|------------------------|------------------------|
> | $\small{\text{LLaMA-2-7B}}$   | $\small{\text{SparseGPT}}$  | $\small{8.98}$         | $\small{9.02}$         | $\small{14.23}$        |
> |                               | $\small{\text{DuoGPT}}$      | $\small{8.58}$         | $\small{8.58}$         | $\small{13.78}$        |
> | $\small{\text{LLaMA-2-13B}}$  | $\small{\text{SparseGPT}}$  | $\small{7.39}$         | $\small{7.41}$         | $\small{9.95}$         |
> |                               | $\small{\text{DuoGPT}}$      | $\small{7.17}$         | $\small{7.18}$         | $\small{9.39}$         |
>
>
> > **Q3: Different LLM architectures**
>
> #### We acknowledge this important concern about different LLM architectures. To address this, we have evaluated DuoGPT across multiple model families including Mistral, Qwen2, and OPT (as shown in our response to **Reviewer B, Q1**). The results demonstrate consistent improvements across all tested architectures, confirming that our theoretical framework and empirical benefits generalize beyond LLaMA models.

---

> > ### Comment · Reviewer_MRdQ · 2025-08-02
> >
> > Thanks for the response; my concerns are carefully addressed.

---

> > > ### Author Response · Authors · 2025-08-02
> > >
> > > Thank you for the positive feedback. We will ensure all your suggestions are reflected in the updated version.

---

### Official Review · Reviewer_f2i7 · 2025-07-02

**Clarity:** 3
**Significance:** 3
**Originality:** 3
**Rating:** 4
**Confidence:** 3

**Summary:**

This paper proposes DuoGPT to improve weight pruning and activation sparsity for large language models. By treating activation sparsity as dynamic structured weight sparsity, DuoGPT constructs a dual-sparse method. Optimal Brain Compression, activation-aware calibration and dense-model output residuals are used for accuracy preservation. Experiments demonstrate that DuoGPT outperforms SOTA structured pruning methods, by up to 9.17% higher accuracy at an iso-speedup of 1.39X.

**Questions:**

- **More models are needed in the experiment:**
  The current experimental evaluation is limited to the LLaMA architecture. It remains unclear how the proposed approach would perform on other model, such as Mistral, Qwen, DeepSeek, etc.

- **Lack of latency measurements:**
  While DuoGPT has dual sparsity from both activations and model weights, this paper does not report latency results. It will be great to understand how latency is affected as well, particularly under varying batch sizes (both large and small). Including latency measurements would strengthen this work.

**Ethical Concerns:**

["NO or VERY MINOR ethics concerns only"]

**Limitations:**

yes

**Quality:**

3

**Strengths And Weaknesses:**

## Strengths

- **Good novelty:**
  Introducing activation sparsity into model pruning is a novel approach. With tailor-designed techniques, DuoGPT successfully improved model pruning performances.

- **Good performance:**
  The method outperforms the SOTA by up to 9.17% accuracy with an iso-speedup of 1.4×.

- **Good experiments:**
  The paper provides comprehensive experiments, including multiple baselines and various evaluation benchmarks.


## Weaknesses

- **No GPU kernel implementation:**
  The paper does not provide a GPU kernel implementation, which make it unclear about the efficiency performance, such as latency, in real-world deployment.

---

> ### Author Rebuttal · Authors · 2025-07-31
>
> #### We thank the reviewer for the valuable comments and suggestions on additional experiments. We have tried our best to address all of them in our rebuttal.
>
> > **Q1: The current experimental evaluation is limited to the LLaMA architecture.**
>
> #### Thank you for this important suggestion. We agree that demonstrating generalizability across architectures is crucial. We have evaluated DuoGPT on multiple model families beyond LLaMA, showing consistent improvements across different architectures. We tested on Mistral, Qwen2, and OPT with varying model size in the table below with 50\% dual-sparsity on WikiText2 PPL.
>
> | $\small{\text{Method}}$   | $\small{\text{Mist-7B}}$ | $\small{\text{Qwen2-7B}}$ | $\small{\text{Qwen2-1.5B}}$ | $\small{\text{OPT-125M}}$ | $\small{\text{OPT-1.3B}}$ |
> |---------------------------|---------------------------|-----------------------------|-------------------------------|----------------------------|----------------------------|
> | $\small{\text{Dense}}$     | $\small{5.25}$           | $\small{7.13}$              | $\small{9.54}$                | $\small{27.65}$            | $\small{14.62}$            |
> | $\small{\text{SparseGPT}}$ | $\small{8.40}$           | $\small{10.91}$             | $\small{19.50}$               | $\small{55.08}$            | $\small{35.25}$            |
> | $\small{\text{DuoGPT}}$    | $\small{{7.91}}$  | $\small{{10.76}}$    | $\small{{18.11}}$      | $\small{{51.88}}$   | $\small{{31.70}}$   |
>
> > **Q2: Latency results.**
>
> #### Thank you for this important suggestion as wel. We provide latency analysis below, noting that our method specifically targets single-batch decoding scenarios where activation sparsity can be effectively exploited through GEMV operations—the critical use case for interactive applications.
>
> #### In the table below, we measured end-to-end latency (% of reduction compared to dense baselines) across different context lengths using the setup described in Appendix B.2, showing consistent improvements that increase with model size (24.6\% average for 7B vs 29.2\% for 13B). This scaling behavior reflects the memory-bound nature of single-batch decoding, where our dual-sparsity approach effectively reduces SRAM loading costs.
>
> | $\small{\text{Model}}$         | $\small{\text{ctxt-50}}$ | $\small{\text{ctxt-100}}$ | $\small{\text{ctxt-200}}$ | $\small{\text{ctxt-500}}$ | $\small{\text{Average}}$ |
> |-------------------------------|---------------------------|----------------------------|----------------------------|----------------------------|---------------------------|
> | $\small{\text{LLaMA-2-7B}}$   | $\small{24.5}$ %           | $\small{25.0}$ %            | $\small{25.0}$ %            | $\small{23.7}$ %            | $\small{24.6}$ %           |
> | $\small{\text{LLaMA-2-13B}}$  | $\small{29.1}$ %           | $\small{28.1}$ %            | $\small{31.5}$ %            | $\small{28.0}$ %            | $\small{29.2}$ %           |
>
> #### We note that significant system optimization opportunities exist for dual-sparse LLM workloads, and we hope this work opens promising avenues for the ML-systems community to explore.
>
> > **Q3: GPU kernel implementation.**
>
> #### Thank you for bringing up this important implementation concern. We want to clarify our current kernel support and future work directions:
>
> #### We have implemented end-to-end triton-based GPU kernels for exploiting activation sparsity, which enables the latency measurements reported in our above response and the throughput results in the main paper. The key idea of the kernel is to use `triton.language.load` to selectively load weight rows based on a bitmask representing the activation sparsity pattern. This demonstrates a lower bound of the real-world efficiency gains from our dual-sparsity approach.
>
> #### The remaining kernel to implement is for compressed storage/loading of unstructured sparse weights. However, this weight compression kernel is orthogonal to our current hardware performance results, as the primary SRAM loading efficiency gains come from the activation sparsity exploitation during GEMV operations.
>
> #### Full kernel support for weight compression represents promising future work that would provide additional HBM loading reduction benefits on top of our current demonstrated improvements. We view this as an exciting direction that could further amplify the efficiency gains of dual-sparse LLM workloads.

---

> > ### Author Response · Authors · 2025-08-06
> >
> > #### Dear Reviewer,
> >
> > #### Thank you again for your detailed review and constructive feedback. In our rebuttal-replies, we have provided additional experiments regarding different models and latency measurements. We also add explanation to our current GPU kernel implementation and the future work direction on the GPU kernel for higher performance gain.
> >
> > #### We would greatly welcome any follow-up questions or comments you might have.
> >
> > #### Best regards,
> >
> > #### Authors

---

> > ### Comment · Reviewer_f2i7 · 2025-08-07
> >
> > Thank you so much to the authors for the rebuttal! I appreciate your responses and I feel this paper is novel and solid.
> >
> > Authors have mostly addressed my concerns with new model variants and latency analysis. I'd like to keep my original positive rating. Thanks again!

---

> > > ### Author Response · Authors · 2025-08-07
> > >
> > > #### Thank you for the reply and the positive feedback! We will ensure all the extra results are added to the updated version. Thanks again!

---

### Official Review · Reviewer_nwtw · 2025-07-03

**Clarity:** 3
**Significance:** 2
**Originality:** 2
**Rating:** 4
**Confidence:** 3

**Summary:**

This paper proposes DuoGPT, a compression method for LLMs that combines structured pruning and unstructured pruning using activation sparsity and weight pruning. It extend the OBC framework with residual correction to accommodate attention sparsity mechanisms. Experimental results demonstrate speedup over dense LLMs, with comparisons against baselines.

**Questions:**

See weaknesses above.

**Ethical Concerns:**

["NO or VERY MINOR ethics concerns only"]

**Final Justification:**

The author’s response addressed most of the concerns.

**Limitations:**

yes

**Quality:**

2

**Strengths And Weaknesses:**

**Strengths**

- The paper proposes a framework that combines structured and unstructured pruning to balance performance and efficiency. It includes adaptive modifications to prior methods like OBC, which is practical.
- Experiments demonstrates the effectiveness and applicability of DuoGPT.

**Weaknesses**

- The method of collecting activation statistics from calibration data and dynamically activating components during decoding is similar to CATS [1], which is also acknowledged in the paper and uses a threshold-based activation sparsity mechanism. A more detailed comparison in both methodology and performance may improve the clarity and originality of this work.
- Why not use ReLU for activation sparsity, which offers a more straightforward and efficient way to reduce computation?
- Combining structured and unstructured pruning seems simple and has been explored in prior works such as STUN [2]. Moreover, the motivation for introducing activation sparsity within the structured pruning component is unclear, since many structured pruning methods are already compatible with unstructured sparsity.
- Some core results (e.g., Table 1, LLaMA-2-70B) show only a marginal improvement in perplexity (e.g., 0.07 better than SparseGPT), while requiring significantly more compute (e.g., +0.6 GPU hours for calibration). The effectiveness of the method seems heavily dependent on additional computation.
- The comparison with SparseGPT does not isolate the effect of residual correction to OBC, as DuoGPT also includes activation sparsity. A separate ablation would help clarify its contribution.

[1] CATS: Contextually-Aware Thresholding for Sparsity in Large Language Models, COLM 2024

[2] STUN: Structured-Then-Unstructured Pruning for Scalable MoE Pruning, arxiv 2024.9

---

> ### Author Rebuttal · Authors · 2025-07-31
>
> #### We thank the reviewer for the valuable comments and insightful suggestions and have tried our best to address all of them in our rebuttal.
>
> > **Q1: Comparison with CATS**
>
> #### Thank you for the suggestion. We acknowledge that CATS is a pioneering work on activation sparsity in LLMs and provide a detailed comparison below.
>
> #### **Methodological Differences:**
> #### CATS focuses on *output*-activation sparsity in gated MLPs by applying thresholds after non-linear activation functions (e.g., zeroing $\text{SiLU}(xW_{\text{gate}})$ neurons below a threshold). This approach targets SRAM savings specifically for $W_{\text{up}}$ and $W_{\text{down}}$.
> #### In contrast, DuoGPT focuses on *input*-activation sparsity by thresholding the *inputs* to each layer. This design allows activation sparsity to be applied across **all** linear layers in LLMs, not just FFN blocks.
>
> #### **Performance Comparison:**
> #### We evaluate both methods on five downstream tasks using LLaMA-2-7B at 40% global sparsity. DuoGPT consistently outperforms CATS across all tasks, as shown below.
>
> | $\small{\text{Method}}$       | $\small{\text{PIQA}}$  | $\small{\text{HSwag}}$ | $\small{\text{ARC-E}}$ | $\small{\text{ARC-C}}$ | $\small{\text{WinoG}}$ | $\small{\text{Avg}}$   |
> |------------------------------|------------------------|------------------------|------------------------|------------------------|------------------------|------------------------|
> | $\small{\text{CATS}}$     | $\small{66.27}$        | $\small{38.48}$        | $\small{45.66}$        | $\small{28.16}$        | $\small{57.38}$        | $\small{47.19}$        |
> | $\small{\text{DuoGPT}}$   | $\small{76.66}$        | $\small{53.42}$        | $\small{53.56}$        | $\small{40.78}$        | $\small{67.17}$        | $\small{58.32}$        |
>
> > **Q2: Why not ReLU?**
>
> #### Thank you for the question. While ReLU can indeed induce activation sparsity, our approach offers several key advantages:
>
> #### 1. ReLU provides output-activation sparsity only for FFN blocks, while our input-activation sparsity works across all linear layers in LLMs.
>
> #### 2. Our method requires no fine-tuning, operating in a one-shot calibration fashion. ReLU replacement requires extensive fine-tuning (e.g., 30B tokens for LLaMA-7B **[1]**).
>
> #### We further compare DuoGPT with the ReLU-based method [1] at 22% global sparsity on LLaMA-2-7B below.
>
> | $\small{\text{Method}}$           | $\small{\text{PIQA}}$  | $\small{\text{HSwag}}$ | $\small{\text{ARC-E}}$ | $\small{\text{ARC-C}}$ | $\small{\text{WinoG}}$ | $\small{\text{Avg}}$   |
> |----------------------------------|------------------------|------------------------|------------------------|------------------------|------------------------|------------------------|
> | $\small{\text{ReLUfication}}$ | $\small{54.08}$        | $\small{25.86}$        | $\small{27.95}$        | $\small{24.06}$        | $\small{48.93}$        | $\small{36.18}$        |
> | $\small{\text{DuoGPT}}$       | $\small{{78.73}}$ | $\small{{56.80}}$ | $\small{{76.47}}$ | $\small{{43.26}}$ | $\small{{67.96}}$ | $\small{{64.64}}$ |
>
> > **Q3: Comparison with structured + unstructured pruning**
>
> #### Thank you for pointing out the STUN reference—we will include it in the related work section. However, we would like to clarify a key misunderstanding: **DuoGPT does *not* combine structured and unstructured pruning.**
>
> #### **Key Methodological Distinction**:
> #### - **STUN** combines actual *structured pruning* (e.g., 5% for non-MoE models) with *unstructured pruning* (often >50%). While this achieves good accuracy, the unstructured component remains difficult to accelerate efficiently on SIMT architectures.
> #### - **DuoGPT**, by contrast, performs only *activation-aware unstructured weight pruning* during calibration. At inference time, we reinterpret activation sparsity as a form of *dynamic structured sparsity* during GEMV operations. This allows for GPU acceleration without the accuracy degradation typically seen in actual structured weight pruning.
>
> #### **Core Innovation**:
> #### We leverage the insight that, in single-batch decoding (GEMV), activation sparsity naturally behaves like structured sparsity: only the weight rows corresponding to non-zero activations are accessed. Building on this, our main technical contributions include extending the OBC framework with a novel closed-form solution that rigorously accounts for activation sparsity effects using asymmetric residual correction (Eq. 3), and develop an efficient GPU implementation that scales to 70B+ models in under 140 minutes.
>
> #### **Performance Comparison**:
> #### On 5-shot GSM8K with LLaMA-2-7B, DuoGPT achieves **comparable accuracy** to STUN (e.g., 7.2% with 47.5% weight sparsity and 15% activation sparsity) while offering **3× greater acceleratable portion** (15% activation sparsity vs. 5% structured sparsity in STUN), demonstrating the practical effectiveness of our dynamic structured sparsity approach.
>
> > **Q4: Dependent on additional computation**
>
> #### We appreciate this concern and would like to clarify two important points:
>
> #### 1. **No Causality Between Compute and Effectiveness**:
> #### One good example would be that the additional calibration time is a constant overhead, independent of the sparsity level. Meanwhile, the effectiveness of DuoGPT increases with dual-sparsity level. As shown in Table 6, our perplexity (PPL) improvement on LLaMA-2-7B grows from **0.5% (at 30% dual-sparsity)** to **12.5% (at 65% dual-sparsity)**—all at the same calibration cost.
>
> #### 2. **Expected Behavior for Large Models**:
> #### Larger models are known to be more resilient to compression. For example, Wanda **[2]** reports PPL improvement dropping from 0.09 (7B) to 0.00 (70B) compared to SparseGPT. Our observed improvement of 0.07 on LLaMA-2-70B is thus reasonable and consistent with prior work.
>
> #### Importantly, the calibration cost is a **one-time offline expense** that scales sublinearly with model size. It is **negligible compared to training costs** and enables efficient inference over the model’s entire deployment lifecycle.
>
> > **Q5: Isolate the effect of residual correction without activation sparsity**
>
> #### Thank you for the suggestion. We conducted the requested ablation to isolate the effect of the residual correction effect by evaluating two scenarios: **(1) weight-only pruning** (no activation sparsity), and **(2) dual-sparsity** (with activation sparsity).
>
> #### The results demonstrate the following:
>
> #### 1. **Residual correction is effective on its own**: DuoGPT outperforms SparseGPT even in the weight-only setting, showing that the second term in Δw (Eq. 3) provides a clear benefit.
>
> #### 2. **Activation sparsity amplifies the gain**: When activation sparsity is included, the magnitude of improvement increases by **2.9×, 1.8×, and 2.6×** respectively across the three models.
>
> #### These results confirm that our closed-form solution adapts effectively: the residual correction term compensates for calibration error from both weight pruning and activation sparsity.
>
> #### The table below shows the results of the ablation study isolating residual correction effects (50% sparsity, WikiText2 PPL)
>
> | $\small{\text{Scenario}}$     | $\small{\text{Method}}$   | $\small{\text{LLaMA2-7B}}$       | $\small{\text{LLaMA2-13B}}$      | $\small{\text{LLaMA3-8B}}$       |
> |------------------------------|---------------------------|-------------------------------|-------------------------------|-------------------------------|
> | $\small{\text{Weight-only}}$ | $\small{\text{SparseGPT}}$ | $\small{7.11}$                | $\small{6.15}$                | $\small{9.98}$                |
> |                              | $\small{\textnormal{DuoGPT}}$    | $\small{6.97\ (\downarrow 0.14)}$ | $\small{6.03\ (\downarrow 0.12)}$ | $\small{9.73\ (\downarrow 0.25)}$ |
> | $\small{\text{Dual-sparsity}}$ | $\small{\text{SparseGPT}}$ | $\small{8.98}$                | $\small{7.39}$                | $\small{14.05}$               |
> |                              | $\small{\text{DuoGPT}}$    | $\small{8.58\ (\downarrow 0.40)}$ | $\small{7.17\ (\downarrow 0.22)}$ | $\small{13.41\ (\downarrow 0.64)}$ |
>
> > **References**
>
> #### **[1]** Mirzadeh, Iman, et al. *"ReLU strikes back: Exploiting activation sparsity in large language models."* arXiv preprint arXiv:2310.04564 (2023)
>
> #### **[2]** Sun, Mingjie, et al. *"A simple and effective pruning approach for large language models."* arXiv preprint arXiv:2306.11695 (2023).

---

> > ### Author Response · Authors · 2025-08-06
> >
> > #### Dear Reviewer,
> >
> > #### Thank you again for your detailed review. We believe our rebuttal has addressed your concerns carefully, particularly regarding the fundamental distinction between our approach and methods like STUN. To clarify: **our method does not combine structured and unstructured pruning**—there is no actual structured pruning. Instead, we reinterpret activation sparsity as dynamic structured weight sparsity, achieving better performance at iso-GPU-speedup compared to actual structured pruning.
> >
> > #### In addition to our Q4 response regarding effectiveness and computation, we respectfully direct you to our response to **Reviewer C, Q1**, where we provide an error bound proof demonstrating our method's guaranteed loss improvement over SparseGPT. This improvement scales with activation sparsity $p_x$ without requiring additional computations. We hope this theoretical proof, combined with our original rebuttal-reply and empirical evidence from Tables 3a and 6, addresses your concern.
> >
> > #### We would greatly appreciate any additional feedback you might have, as your insights are valuable to improving our work.
> >
> > #### Best regards,
> >
> > #### Authors

---

> > > ### Comment · Reviewer_nwtw · 2025-08-07
> > >
> > > Thank you for your response. I will raise my score to 4.

---

> > > > ### Author Response · Authors · 2025-08-07
> > > >
> > > > #### Thank you very much for raising the score. We will ensure all the new results are added to the updated version of the paper.

---

### Note · Authors · 2025-08-12

#### We are grateful for the constructive feedback and suggestions from all three reviewers and the opportunity to provide these final comments.


#### **Takeaways from rebuttal discussions:**

#### - Clarified that our approach does not combine structured and unstructured pruning, but rather reinterprets activation sparsity as dynamic structured weight sparsity
#### - Explained the non-causal relationship between the effectiveness of our method and the extra computation
#### - Provided additional experiments across multiple architectures (Mistral, Qwen2, OPT) demonstrating broad applicability
#### - Included latency analysis and GPU kernel implementation explanations
#### - Added theoretical rigor with error bound proofs showing guaranteed improvement over SparseGPT baselines
#### - Demonstrated the compatibility of our method with weight quantization

#### We will incorporate all suggested improvements, additional experimental results, and clarifications into the final manuscript to strengthen the paper.

#### We believe this work opens promising new directions not only for dual-sparse LLM algorithm optimization but also for the ML-systems community to explore system-level optimizations for dual-sparse LLM deployment. We look forward to contributing to the efficient LLM community with our work.

#### Thanks very much again for all the reviewers and ACs for their efforts and supports during the reviewing process.

---

### Decision · Program_Chairs · 2025-09-17

**Decision:**

Accept (poster)

**Comment:**

This paper proposes to combine "dynamic" activation sparsity (which can be seen as per-token structured sparsity) with "static" unstructured weight sparsity. This involves using a Hessian-based correction mechanism (i.e., as in OBC/SparseGPT) that takes the activation sparsity into account in the weight pruning/recalibration stage. The approach is tested on the LLaMA family of models where it is found to result in a better sparsity-performance frontier than baseline methods such as SparseGPT/CATS/TEAL.

This is a well-executed paper that demonstrates solid improvements over strong baselines. The authors also provide a Triton-based implementation to enable actual wallclock speedups. On the negative side, there are some concerns with regard to novelty (in some sense, it is more or less just combining SparseGPT with CATS/TEAL), as well as whether this approach would generalize to other families of models. During the rebuttal phase, the authors showed that this method generalizes to non-LLaMA families as well.

Overall, despite some concerns about novelty, I believe that this is a solid contribution that would be of benefit to the community.